# The molecular basis of coupling between poly(A)-tail length and translational efficiency

**Kehui Xiang**[1,2,3], **David P Bartel**[1,2,3]*

[1]Howard Hughes Medical Institute, Cambridge, United States; [2]Whitehead Institute for Biomedical Research, Cambridge, United States; [3]Department of Biology, Massachusetts Institute of Technology, Cambridge, United States

**Abstract** In animal oocytes and early embryos, mRNA poly(A)-tail length strongly influences translational efficiency (TE), but later in development this coupling between tail length and TE disappears. Here, we elucidate how this coupling is first established and why it disappears. Overexpressing cytoplasmic poly(A)-binding protein (PABPC) in *Xenopus* oocytes specifically improved translation of short-tailed mRNAs, thereby diminishing coupling between tail length and TE. Thus, strong coupling requires limiting PABPC, implying that in coupled systems longer-tail mRNAs better compete for limiting PABPC. In addition to expressing excess PABPC, post-embryonic mammalian cell lines had two other properties that prevented strong coupling: terminal-uridylation-dependent destabilization of mRNAs lacking bound PABPC, and a regulatory regime wherein PABPC contributes minimally to TE. Thus, these results revealed three fundamental mechanistic requirements for coupling and defined the context-dependent functions for PABPC, which promotes TE but not mRNA stability in coupled systems and mRNA stability but not TE in uncoupled systems.

*For correspondence:
dbartel@wi.mit.edu

**Competing interests:** The authors declare that no competing interests exist.

## Introduction

Most eukaryotic mRNAs are polyadenylated at their 3′ ends in a process associated with transcriptional termination. In the nucleus, these poly(A) tails can facilitate mRNA nucleocytoplasmic export (*Kühn and Wahle, 2004*), whereas in the cytoplasm, they serve as molecular timers for mRNA decay, with their lengths becoming progressively shorter by deadenylation, which eventually leads to mRNA de-capping and turnover (*Chen and Shyu, 2011*; *Eisen et al., 2020*; *Goldstrohm and Wickens, 2008*).

The length of a poly(A) tail can also influence mRNA translational efficiency (TE). Pioneering studies in maturing oocytes and early embryos show that lengthening of poly(A) tails through cytoplasmic polyadenylation is critical for regulating gene expression during these early stages of animal development (*Richter, 1999*; *Sallés et al., 1994*; *Sheets et al., 1995*). Results from these and other single-gene studies in oocytes and early embryos had led to the notion that the length of a poly(A) tail generally correlates with TE (*Eckmann et al., 2011*; *Weill et al., 2012*). More recent transcriptome-wide studies confirm a strong global relationship between tail length and TE in oocytes and early embryos (*Eichhorn et al., 2016*; *Lim et al., 2016*; *Subtelny et al., 2014*). However, in fish, frogs, and flies, this correlation diminishes near the time of gastrulation, and coupling between poly(A)-tail length and TE is essentially nonexistent in post-embryonic systems (*Eichhorn et al., 2016*; J.-E. *Park et al., 2016*; *Subtelny et al., 2014*). Thus, these global analyses reveal a developmental transition in how translation is regulated (*Subtelny et al., 2014*), which closely follows the long-known maternal-to-zygotic transition in transcriptional control. The existence of this transition in translational control brings to the fore mechanistic questions as to how coupling between poly(A)-

**eLife digest** Cells are microscopic biological factories that are constantly creating new proteins. To do so, a cell must first convert its master genetic blueprint, the DNA, into strands of messenger RNA or mRNA. These strands are subsequently translated to make proteins. Cells have two ways to adjust the number of proteins they generate so they do not produce too many or too few: by changing how many mRNA molecules are available for translation, and by regulating how efficiently they translate these mRNA molecules into proteins.

In animals, both unfertilized eggs and early-stage embryos lack the ability to create or destroy mRNAs, and consequently cannot adjust the number of mRNA molecules available for translation. These cells can therefore only regulate how efficiently each mRNA is translated. They do this by changing the length of the so-called poly(A) tail at the end of each mRNA molecule, which is made up of a long stretch of repeating adenosine nucleotides. The mRNAs with longer poly(A) tails are translated more efficiently than those with shorter poly(A) tails. However, this difference disappears in older embryos, when both long and short poly(A) tails are translated with equal efficiency, and it is largely unknown why.

To find out more, Xiang and Bartel studied frog eggs, and discovered that artificially raising levels of a protein that binds poly(A) tails, also known as PABPC, improved the translation of short-tailed mRNAs to create a situation in which both short- and long-tailed mRNAs were translated with near-equal efficiency. This suggested that short- and long-tailed mRNAs compete for limited amounts of the translation-enhancing PABPC, and that long-tailed mRNAs are better at it than short-tailed mRNAs. Further investigation revealed that eggs also had to establish the right conditions for PABPC to enhance translation and had to protect mRNAs not associated with PABPC from being destroyed before they could be translated.

Overall, Xiang and Bartel found that in eggs and early embryos, PABPC and poly(A) tails enhanced the translation of mRNAs but did not influence their stability, whereas later in development, they enhanced mRNA stability but not translation.

This research provides new insights into how protein production is controlled at different stages of animal development, from unfertilized eggs to older embryos. Understanding how this process is regulated during normal development is crucial for gaining insights into how it can become dysfunctional and cause disease. These findings may therefore have important implications for research into areas such as infertility, reproductive medicine and rare genetic diseases.

tail length and TE is established in oocytes and early embryos and why this coupling disappears later in development.

Cytoplasmic poly(A)-binding proteins (PABPCs) are highly conserved RNA-binding proteins in eukaryotes (*Mangus et al., 2003*). Although *Saccharomyces cerevisiae* has only one PABPC (Pab1p), most animals contain multiple paralogs that have spatially and temporally varied expression patterns (*Smith et al., 2014*; *Wigington et al., 2014*). PABPCs have high affinity to poly(A) sequences in vitro ($K_d$ ~5 nM for $A_{25}$) and require at least 12 As for efficient binding (*Kühn and Wahle, 2004*). Binding of PABPCs to mRNA poly(A) tails can enhance translation, but the mechanism of this enhancement is unclear. One model posits that the mRNA forms a closed-loop structure mediated by the association of the eukaryotic translation initiation factor eIF4G (a scaffolding protein) with both PABPC and the cap-binding protein eIF4E (*Hinnebusch, 2014*; *Thompson and Gilbert, 2017*; *Wells et al., 1998*). This association is proposed to stabilize the interaction between eIF4E and the mRNA 5′ cap and facilitate recruitment and/or recycling of ribosomes to increase translation initiation (*Kahvejian et al., 2001*). However, despite direct visualization of loop-like assemblies both within some cells and in an in vitro reconstituted system (*Christensen et al., 1987*; *Wells et al., 1998*), results of several studies have questioned the universality of this model among different mRNAs and biological systems (*Adivarahan et al., 2018*; *Amrani et al., 2008*; *Costello et al., 2015*; *Rissland et al., 2017*; *Thompson and Gilbert, 2017*).

PABPCs can also influence mRNA stability, as shown in yeast. Genetic ablation of yeast Pab1p is lethal and causes lengthening of steady-state poly(A)-tail lengths (*Sachs and Davis, 1989*), which is attributed to pre-mature mRNA decapping and compromised deadenylation (*Caponigro and*

*Parker, 1995*). Both yeast and mammalian PABPCs can interact with two mRNA deadenylation complexes PAN2-PAN3 and CCR4-NOT, and either promote or inhibit their activities in vitro (*Schäfer et al., 2019*; *Uchida et al., 2004*; *Webster et al., 2018*; *Yi et al., 2018*). Because mRNA decay is coupled to deadenylation (*Decker and Parker, 1993*; *Eisen et al., 2020*), the deadenylation-stimulatory effects of PABPC would accelerate the demise of bound mRNAs, which contrasts to other studies suggesting PABPC protects mRNAs from degradation in cell extracts (*Bernstein et al., 1989*; *Wang et al., 1999*). The dichotomous and potentially conflicting functions of metazoan PABPC examined in vitro raise the question of the extents to which PABPC might influence mRNA poly(A)-tail length and stability in metazoan cells.

PABPCs are generally thought to coat mRNA poly(A) tails in the cytoplasm (*Kühn and Wahle, 2004*; *Mangus et al., 2003*). However, the stoichiometry between PABPC and poly(A) sites might vary in different biological contexts (*Cosson et al., 2002*; *Voeltz et al., 2001*), and it is unclear whether this potentially variable stoichiometry might impact gene regulation in cells. Moreover, the possibility that PABPC might influence protein synthesis by affecting either mRNA stability or TE can complicate analysis of its molecular functions in different biological systems, leaving its mechanistic roles poorly understood.

Here, we uncover mechanistic requirements for coupling between poly(A)-tail length and TE observed in oocytes and early embryos, showing that this coupling and the subsequent uncoupling observed later in development rely on a context-dependent switch in the function of PABPCs.

## Results

### Limiting PABPC is required for tail length to strongly influence TE of reporter mRNAs

To assay the influence of poly(A)-tail length on TE, we used an in vitro translation extract made from stage VI *Xenopus laevis* oocytes, where cytoplasmic polyadenylation leads to translational activation of the *c-mos*, *cdk2* and some cyclin mRNAs (*Richter and Lasko, 2011*). Into this extract we added *Nanoluc* luciferase (*Nluc*) reporter mRNAs with either a short (29 nt) or a long (139 nt) poly(A) tail (*Figure 1A*). These mRNAs were made by in vitro transcription from DNA templates that encoded the mRNA body followed by the poly(A) tail as well as the hepatitis delta virus (HDV) self-cleaving ribozyme, which cleaved during in vitro transcription to generate not only a defined 3′ end at the desired poly(A)-tail length but also a 2′−3′-cyclic phosphate designed to inhibit undesired lengthening or shortening of the tail (*Avis et al., 2012*).

When added to the frog oocyte extracts together with a firefly luciferase (*Fluc*) mRNA, used to normalize for overall translation activity, the long-tailed reporter was translated substantially better than was the short-tailed reporter (*Figure 1B*). In contrast, the same reporter mRNAs were translated nearly equally well in rabbit reticulocyte lysate, a post-embryonic differentiated system for which no coupling between tail length and TE was expected (*Subtelny et al., 2014*; *Figure 1B*). Similar results were observed for an analogous pair of *Renilla* luciferase reporter mRNAs (*Figure 1—figure supplement 2A*). In both the oocyte and reticulocyte systems, the reporter mRNAs were stable with no detectable changes to their tail lengths (*Figure 1—figure supplement 1A–C*). Thus, the large difference in luciferase signal observed between the short- and long-tailed reporters in the oocyte extract was attributable to a difference in TE. These results showed that the causal relationship between longer poly(A)-tail length and greater TE observed for some maturation-specific mRNAs in frog oocytes (*Sheets et al., 1995*; *Stebbins-Boaz and Richter, 1994*) is not unique to those mRNAs, and indicated that frog oocyte extracts provide a system for probing the mechanism that couples tail length to TE.

When considering the potential mechanisms for reading out tail length and promoting translation, a role for PABPC seemed plausible. For instance, translation might be sensitive to the number of PABPC molecules associated with an mRNA. In one mechanistic possibility, PABPC might be in excess over its binding sites within tails, such that tails are coated with the protein, as is generally thought to occur (*Kühn and Wahle, 2004*; *Mangus et al., 2003*), in which case, mRNAs with longer tails might be detected as those able to bind more PABPC molecules. At another mechanistic extreme, PABPC might be limiting, such that mRNAs compete with each other for PABPC binding, in which case, those with long poly(A)-tail lengths would compete more effectively and thereby

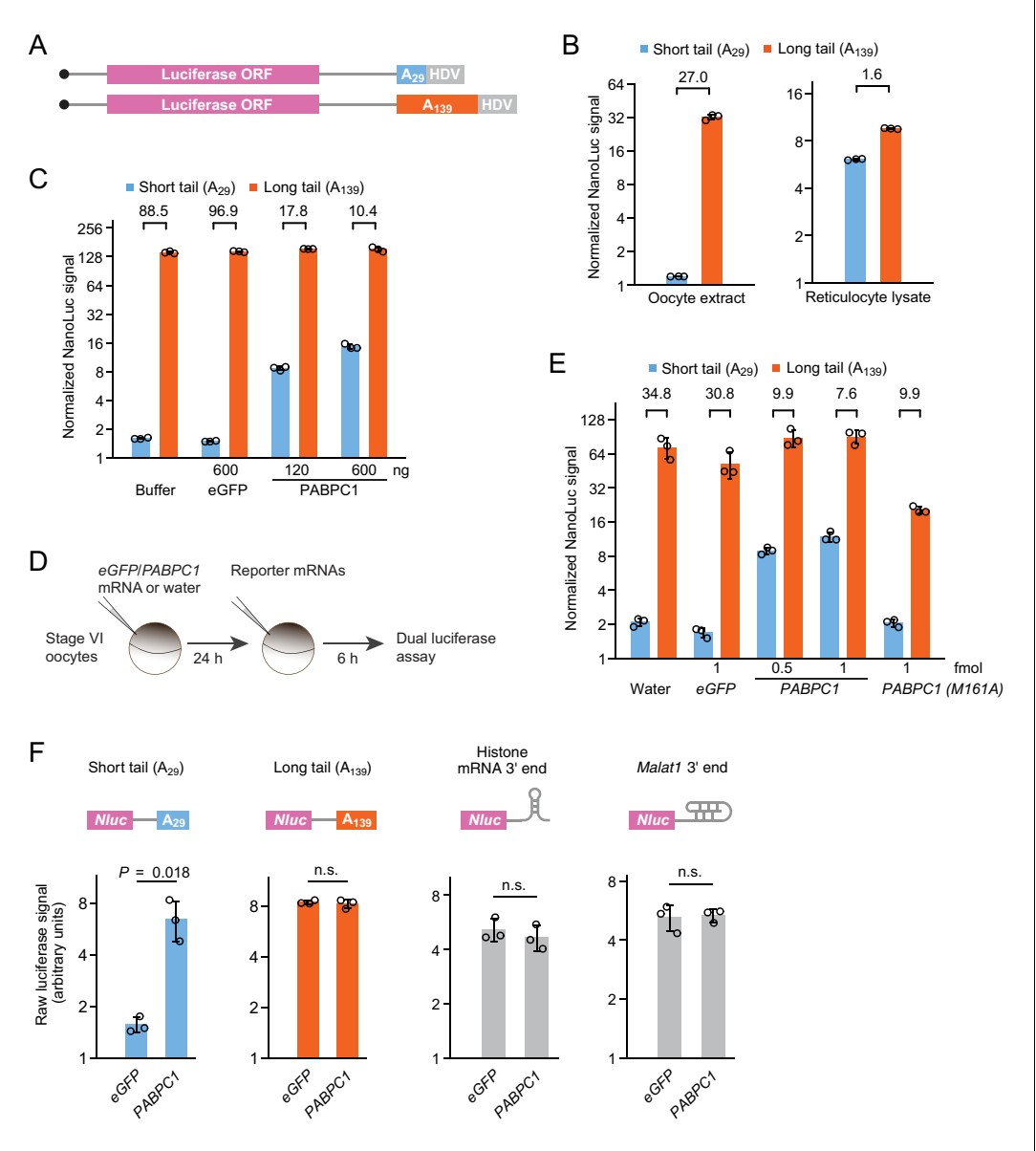

**Figure 1.** PABPC overexpression uncouples poly(A)-tail length and TE in frog oocytes. (A) Schematic of capped T7 transcripts with two different tail lengths, which were used as reporter mRNAs. Additional sequences beyond the HDV sequence are not depicted. (B) The effect of tail length on relative yields of in vitro translation of short- and long-tailed *Nluc* reporter mRNAs, in either frog oocyte extract (left) or rabbit reticulocyte lysate (right). The number above each bracket indicates the fold difference of the mean normalized luciferase signal (error bars, standard deviation from three technical replicates). (C) The effect of purified PABPC1 on relative yields of in vitro translation of short- and long-tailed *Nluc* reporter mRNAs in frog oocyte extract. Purified eGFP and PABPC1 were each added as indicated. Otherwise, this panel is as in (B). (D) Experimental scheme for serial-injection of mRNAs into frog oocytes. (E) The effect of overexpressing PAPBC1 and a PABPC1 M161A mutant on relative translation of *Nluc* reporter mRNAs in frog oocytes. Differential PABPC1 expression was achieved by injecting the indicated amount of mRNA in the first injection (error bars, standard deviation from three biological replicates). Otherwise, this panel is as in (B). (F) The effect of PAPBC1 overexpression on translation of reporter mRNAs with different 3'-end structures in frog oocytes. Shown are raw luciferase yields from *Nluc* reporters that have either a short poly(A) tail, a long poly(A) tail, a histone mRNA 3'-end stem-loop, or a *Malat1* triple-helix 3'-end in oocytes overexpressing either eGFP or PABPC1 (error bars, standard deviations from three biological replicates). p values are from one-sided *t*-tests (n.s., not significant). For overexpression, 2.4 fmol mRNA was injected per oocyte.

The online version of this article includes the following source data and figure supplement(s) for figure 1:

*Figure 1 continued on next page*

*Figure 1 continued*

**Source data 1.** Source data for luciferase values shown in *Figure 1* and *Figure 1—figure supplement 2*.

**Figure supplement 1.** Supporting data for reporter experiments examining the effect of PAPBC levels on coupling between tail length and translation.

**Figure supplement 2.** Additional reporter assays examining the effect of PAPBC levels on coupling between tail length and translation.

---

preferentially benefit from any enhancement in TE that PABPC binding confers. To distinguish between these possibilities, we increased available PABPC in our oocyte extracts, reasoning that if PABPC were already coating the tails, adding more would have little effect, whereas if PABPC were limiting, adding more would diminish the competition for PABPC binding and thereby reduce the difference in TE observed between short- and long-tailed mRNAs. Accordingly, we purified recombinant *Xenopus* PABPC1 to near homogeneity (*Figure 1—figure supplement 1D*) and examined its influence when added to the in vitro translation extract derived from stage VI oocytes. As more PABPC1 was added, translation of the short-tailed reporter increased, with little change in translation of the long-tailed reporter, whereas adding equivalent amount of eGFP had little impact on translation of either reporter (*Figure 1C*). This concentration-dependent diminution of coupling between tail-length and TE strongly supported the hypothesis that limiting PAPBC is required for strong coupling.

To investigate whether this requirement of limiting PABPC was restricted to our in vitro extracts or whether it also applied to living oocytes, we performed serial-injection experiments in oocytes. Stage VI frog oocytes were first injected with either *PABPC1* mRNA or a control, and after waiting 24 hr to allow PABPC1 protein to accumulate (*Figure 1—figure supplement 1E*), oocytes were injected with the reporter mRNAs and assayed for luciferase activity (*Figure 1D*). Whereas injecting the control mRNA, *eGFP*, had no more influence than injecting water, injecting *PABPC1* mRNA significantly reduced the extent to which poly(A)-tail length and TE were coupled (*Figure 1E*). Similar results were observed for an analogous pair of *Renilla* luciferase reporter mRNAs or when injecting *ePAB* mRNA rather than *PABPC1* mRNA (*Figure 1—figure supplement 2B*). Reporter poly(A)-tail lengths did not change over the course of the experiment (*Figure 1—figure supplement 1F*), which indicated that the increased relative translation of the short-tailed reporter mRNA was not due to elongated poly(A) tails.

Introducing additional PABPC into frog oocytes specifically improved translation of the short-tailed reporter while having little effect on translation of either the long-tailed reporter or reporters for which tails were replaced with either a stem-loop from the 3′ end of a histone mRNA (*Ling et al., 2002*) or a triple-helix from the 3′ end of the *Malat1* non-coding RNA (*Wilusz et al., 2012*; *Figure 1F*). The observation that mRNAs required a tail to benefit from added PABPC indicated that the effects of adding PABPC were mediated in cis through tail-bound PABPC molecules, and were direct and not some secondary consequence of altering translation. Moreover, the observation that PABPC had little effect on translation of long-tailed mRNAs suggested that these mRNAs competed for the limiting endogenous PABPC so effectively that binding of additional PABPC imparted no detectable additional benefit to their translation.

Introducing *PABPC1(M161A)*, which encodes a PABPC1 mutant that is unable to bind eIF4G (*Groft and Burley, 2002*), also diminished coupling but did so by repressing translation of the long-tailed reporter. The reduced translation of the long-tailed reporter was presumably due to a dominant-negative effect of replacing functional endogenous PABPC molecules with defective ones. The observation that the long-tailed reporter was preferentially affected agreed with our conclusion that endogenous PABPC was limiting and preferentially binding to long-tailed mRNAs. The idea that the M161A mutant was unable to enhance translation in frog oocytes implied that the ability for PABPC to bind eIF4G and form the closed-loop structure is important for enhancing translation in this context (*Wakiyama et al., 2000*).

In summary, our results with reporters in oocytes and oocyte extracts confirmed both the positive effect of PABPC on translation and the causal relationship between poly(A)-tail length and TE in these systems. Moreover, these results revealed that strong coupling between poly(A)-tail length and TE requires limiting PABPC.

## Limiting PABPC is required for tail length to strongly influence TE of endogenous mRNAs

To examine the global effect of increasing PABPC on the translational regulatory regime acting in the oocyte, we monitored the relationship between tail length and TE for endogenous mRNAs of the oocytes. As expected from results of single-gene experiments in frog oocytes (*Figure 1E*; *Sheets et al., 1995*; *Stebbins-Boaz and Richter, 1994*) and the strong coupling between poly(A)-tail length and TE observed in both frog embryos and fly oocytes (*Eichhorn et al., 2016*; *Lim et al., 2016*; *Subtelny et al., 2014*), we found that poly(A)-tail length correlated strongly with TE in stage VI frog oocytes (*Figure 2A*). Overexpressing either PABPC1 or ePAB in these oocytes significantly diminished the coupling, with the Spearman correlation ($R_s$) for the relationship between tail length and TE dropping from 0.62 to 0.36 and 0.38, respectively (*Figure 2A*, both p = 0, modified Dunn and Clark's z-test [*Diedenhofen and Musch, 2015*]). In contrast, overexpressing eGFP had no significant impact on the coupling (p = 0.11), which indicated that this transcriptome-wide effect was a result of additional PABPC protein rather than a non-specific effect of adding more mRNA.

Accompanying the reduced coupling observed upon PABPC overexpression was a significant relative increase of TE for short-tailed mRNAs, an effect not observed in eGFP-expressing oocytes (*Figure 2—figure supplement 1A*). This TE increase was not accompanied by corresponding lengthening of poly(A) tails (*Figure 2—figure supplement 1B*), implying that tail-length changes did not cause these relative TE changes. To make comparisons of absolute TE changes, we repeated the ePAB-overexpression experiment but omitted rRNA depletion during sequencing library construction, thereby allowing us to normalize TE using mitochondrial mRNAs (*Iwasaki et al., 2016*), which were otherwise depleted by Illumina Ribo-Zero kits (*Figure 2—figure supplement 1C*). In this experiment, we also injected oocytes with a short-tailed *Nluc* mRNA reporter and a long-tailed *Fluc* mRNA reporter and monitored their absolute TE changes together with those of endogenous mRNAs. Most endogenous mRNAs had greater absolute TE in ePAB-overexpressing oocytes compared to eGFP-expressing control oocytes (*Figure 2B*). This result was consistent with [35]S metabolic-labeling experiments showing that overexpression of PABPC1 but not eGFP significantly increased global protein synthesis in oocytes (*Figure 2C–D*). Moreover, the magnitude of the TE increase conferred by ePAB-overexpression negatively correlated with tail length, which showed that translation of short-tailed mRNAs improved substantially more than that of long-tailed mRNAs (*Figure 2E*), as observed for our co-injected reporters. Indeed, adding ePAB had essentially no overall effect on TE of endogenous mRNAs with the longest tails (median TE fold change = 1.06 for the 54 mRNAs with median tail lengths > 80 nt), as observed for our long-tailed reporters. The preferential improvement of TEs for short-tailed mRNAs led to not only an overall shift in TE but also narrowing of the TE distribution (*Figure 2F*) to more closely resemble the distributions observed in cells in which poly(A)-tail length and TE are not coupled (*Subtelny et al., 2014*). These results supported the hypothesis that increasing PABPC in oocytes increases the opportunity for short-tailed mRNAs to bind a PABPC molecule, thereby promoting translation.

Overall, the results of our global analyses of mRNAs in frog oocytes agreed with those of reporter assays, thereby extending to endogenous mRNAs support for the conclusion that limiting PABPC plays a critical role in conferring strong coupling between poly(A)-tail length and TE.

## Intragenic analyses further demonstrate the importance of limiting PABPC for establishing coupling between tail length and TE

Our global analysis examining the relationship between poly(A)-tail length and TE of endogenous mRNAs in oocytes differed from our reporter assays in that the comparison was made between mRNAs of different genes, which can be confounded by features other than tail length that vary between these mRNAs. To overcome this issue, we developed a high-throughput method for comparing effects on different tail-length isoforms from each gene. This approach for intragenic analyses, called PAL-TRAP (*P*oly(*A*) tail-*L*ength profiling following *T*ranslating *R*ibosome *A*ffinity *P*urification), resembled other TRAP approaches in that ribosomes were sparsely tagged such that their immunoprecipitation (IP) preferentially isolated mRNA isoforms associated with more ribosomes, which were inferred to be more highly translated (*Chen and Dickman, 2017*; *Heiman et al., 2008*). In a system in which poly(A)-tail length and TE were coupled, longer-tail mRNAs were

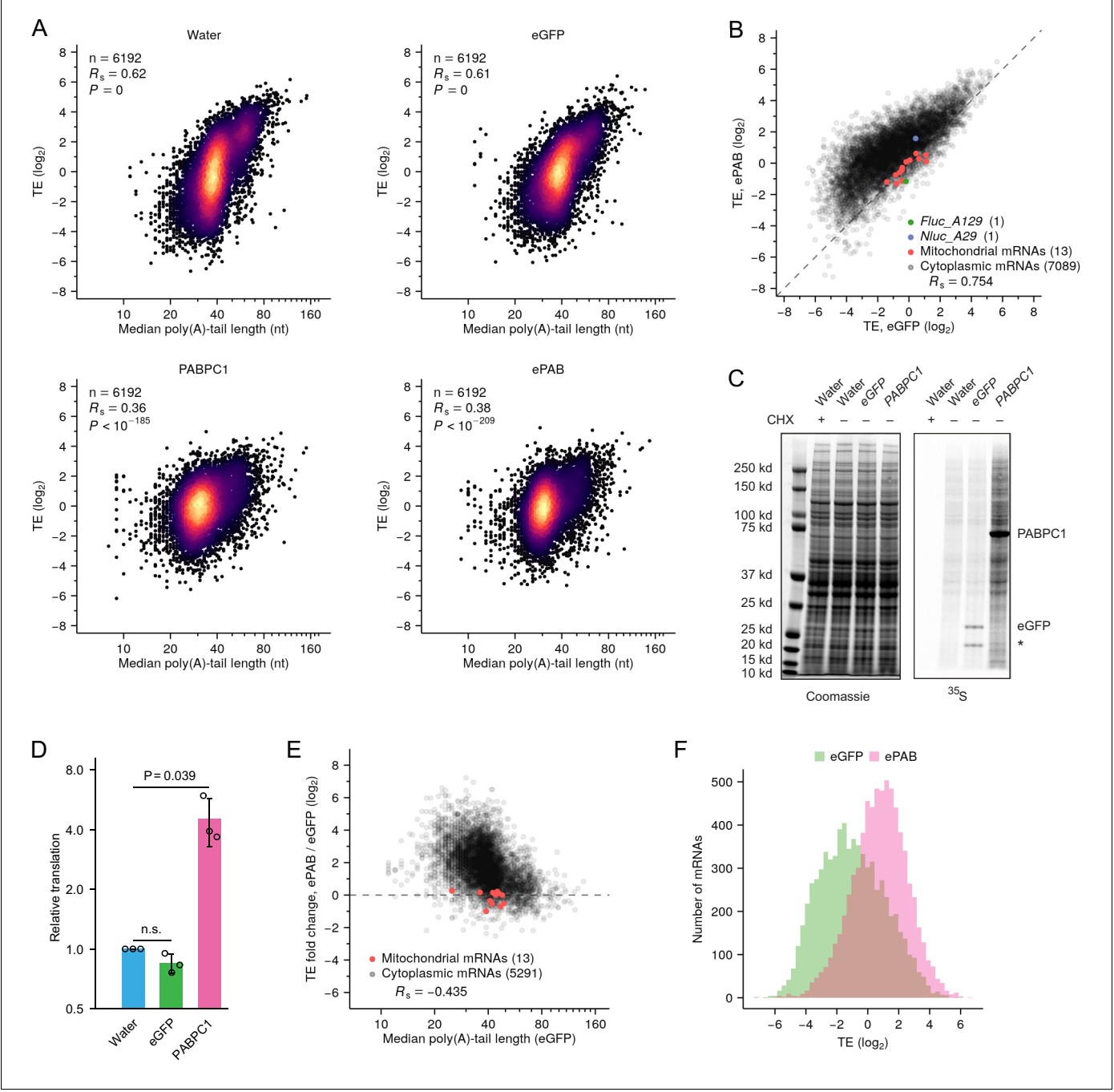

**Figure 2.** Increased PABPC promotes translation of endogenous short-tailed mRNAs, thereby diminishing coupling between tail length and TE. (**A**) The effect of PABPC on coupling between tail length and TE in frog oocytes. Shown is the relationship between TE and median poly(A)-tail length in oocytes injected with either water or mRNAs encoding either eGFP, PABPC1, or ePAB (injecting 16 fmol mRNA per oocyte). Results are shown for mRNAs from genes with ≥100 poly(A) tags. Each poly(A) tag represents a pair of sequencing reads that identify the mRNA and its poly(A) tail length. $R_s$ is the Spearman's correlation coefficient. (**B**) A global increase in TE observed upon overexpressing ePAB. TE in ePAB-overexpressing oocytes is compared with TE in eGFP-expressing oocytes. Experiment was as in A, except rRNAs were not depleted, and TE values are normalized to those of mitochondrial mRNAs. Also shown are results from a co-injected short-tailed *Nluc* reporter and a long-tailed *Fluc* reporter. (**C**) Effect of expressing PABPC1 on protein synthesis in oocytes. At the left is a gel showing total protein after injection of either water or the indicated mRNA (injecting 4 fmol mRNA per oocyte), with or without treatment with cycloheximide (CHX), as visualized by Coomassie staining. At the right is the same gel showing protein synthesis, as visualized by incorporation of $^{35}$S-methionine and $^{35}$S-cysteine. Prominent bands presumably represent PABPC1, eGFP, and a not fully denatured form of eGFP (asterisk) expressed from injected mRNAs. (**D**) Quantification of the effect of expressing PABPC1 on protein synthesis in oocytes, as measured in (**C**) and two additional biological replicates. Only regions above the PABPC1 band, and between the PABPC1 band and the

*Figure 2 continued on next page*

*Figure 2 continued*

top eGFP band, were used for quantification. Values were normalized to that of the mean value from water-injected oocytes (error bars, standard deviation; p values, one-sided *t*-tests; n.s., not significant). (E) The preferential effect of overexpressing ePAB on the TE of short-tailed mRNAs. TE fold changes observed between ePAB-overexpressing and eGFP-expressing oocytes are plotted as a function of median tail length in eGFP-expressing oocytes. TE and tail-length values were obtained from different batches of oocytes; results are shown for mRNAs from genes with ≥100 poly(A) tags. TE values are normalized to those of mitochondrial mRNAs. (F) Effect of overexpressing ePAB on the distribution of TE values observed in frog oocytes. Shown is the TE distribution observed in ePAB-overexpressing oocytes and that observed in eGFP-expressing oocytes. Tail-length measurements in this figure were obtained using TAIL-seq.

The online version of this article includes the following source data and figure supplement(s) for figure 2:

**Source data 1.** Source data for values shown in *Figure 2D*.
**Figure supplement 1.** Increased PABPC increases translation of endogenous short-tailed mRNAs in frog oocytes.

expected to be associated with more ribosomes and therefore enriched in the eluate (*Figure 3A*), whereas in an uncoupled system, longer-tail mRNAs were not expected to be enriched in the eluate.

To implement PAL-TRAP, we first injected stage VI frog oocytes with an mRNA encoding C-terminal HA-tagged RPL3 (*Chen and Dickman, 2017*) and allowed time for RPL3-HA protein expression and incorporation into ribosomes (*Figure 3B*). As a control, HA-tagged eGFP was expressed in separate oocytes. To examine the requirement of limiting PABPC for coupling between poly(A)-tail length and TE, we then injected oocytes with mRNAs that expressed either eGFP or ePAB. Confirming that RPL3-HA was incorporated into functional ribosomes, HA-IP from RPL3-HA-expressing oocytes enriched for proteins from both ribosomal subunits, cytoplasmic mRNAs, and ePAB, whereas HA-IP from eGFP-HA-expressing oocytes did not (*Figure 3—figure supplement 1A–C*). In control oocytes expressing eGFP, longer-tail mRNAs were enriched in the eluate compared to the input (median tail lengths 41 and 36 nt, respectively), which reflected the coupling between tail length and TE (*Figure 3C*). Overexpression of ePAB reduced this enrichment for longer-tail mRNAs in the eluate compared to the input (median tail lengths 37 and 35 nt, respectively), as expected if limiting PABPC was required for this coupling (*Figure 3C*). We also analyzed the flowthrough fractions from these pulldown experiments. For both eGFP- and ePAB-expressing oocytes, the cumulative distribution of poly(A)-tail lengths from the flowthrough was nearly identical to that from the input (*Figure 3—figure supplement 1D*), as expected when considering that only a small fraction of input was depleted by HA-beads. These analyses indicated that our methods were able to capture small differences in tail-length distributions.

When analyzing, for each gene, mRNAs with different tail-length isoforms, mRNAs from most genes (84.4%) had longer median poly(A)-tail lengths in the eluate than in the input (*Figure 3D–E*), implying that for mRNAs from most genes, long-tailed isoforms were translated more efficiently than short-tailed ones. Although the median of the median tail-length differences was moderate (+2.5 nt), 28.3% mRNAs had a ≥ 5 nt longer median tail length in the eluate (*Figure 3E*). In contrast, when input and flowthrough were compared, the median tail-length differences centered on 0 nt, and mRNAs from only 5.4% of genes had a ≥ 5 nt longer median tail length in one of the samples, as expected when considering that only a small fraction of input was depleted by HA-beads (*Figure 3—figure supplement 1E–F*). Overexpressing ePAB reduced the number of genes with long-tailed isoforms enriched in the eluate and shifted the distribution of median tail-length differences closer to 0 nt (*Figure 3D–E*), as expected if coupling between poly(A)-tail length and TE diminished.

In these PAL-TRAP experiments, a mixture of *Rluc* reporter mRNA molecules with different poly(A)-tail lengths was co-injected with mRNAs that expressed either eGFP or ePAB. In eGFP-expressing oocytes, longer-tail *Rluc* isoforms were highly enriched in the eluate compared to the input, whereas in ePAB-overexpressing oocytes, this difference diminished dramatically (*Figure 3F*). As expected, *Nluc* reporter mRNA, which was added to the lysate as a spike-in during HA pulldown, was not significantly enriched in longer-tail species in the eluate, regardless of the treatment, which indicated that the changes observed for *Rluc* mRNA reflected changes occurring in the oocyte, prior to lysis (*Figure 3F*). Although endogenous mRNAs from some genes, such as *btg4.S*, also underwent large changes in median tail differences in response to ePAB-overexpression (*Figure 3F*), changes were typically smaller for endogenous mRNAs (*Figure 3E*), which was at least partly attributable to the narrower range of initial tail length isoforms for endogenous mRNAs compared to the injected *Rluc* mRNA.

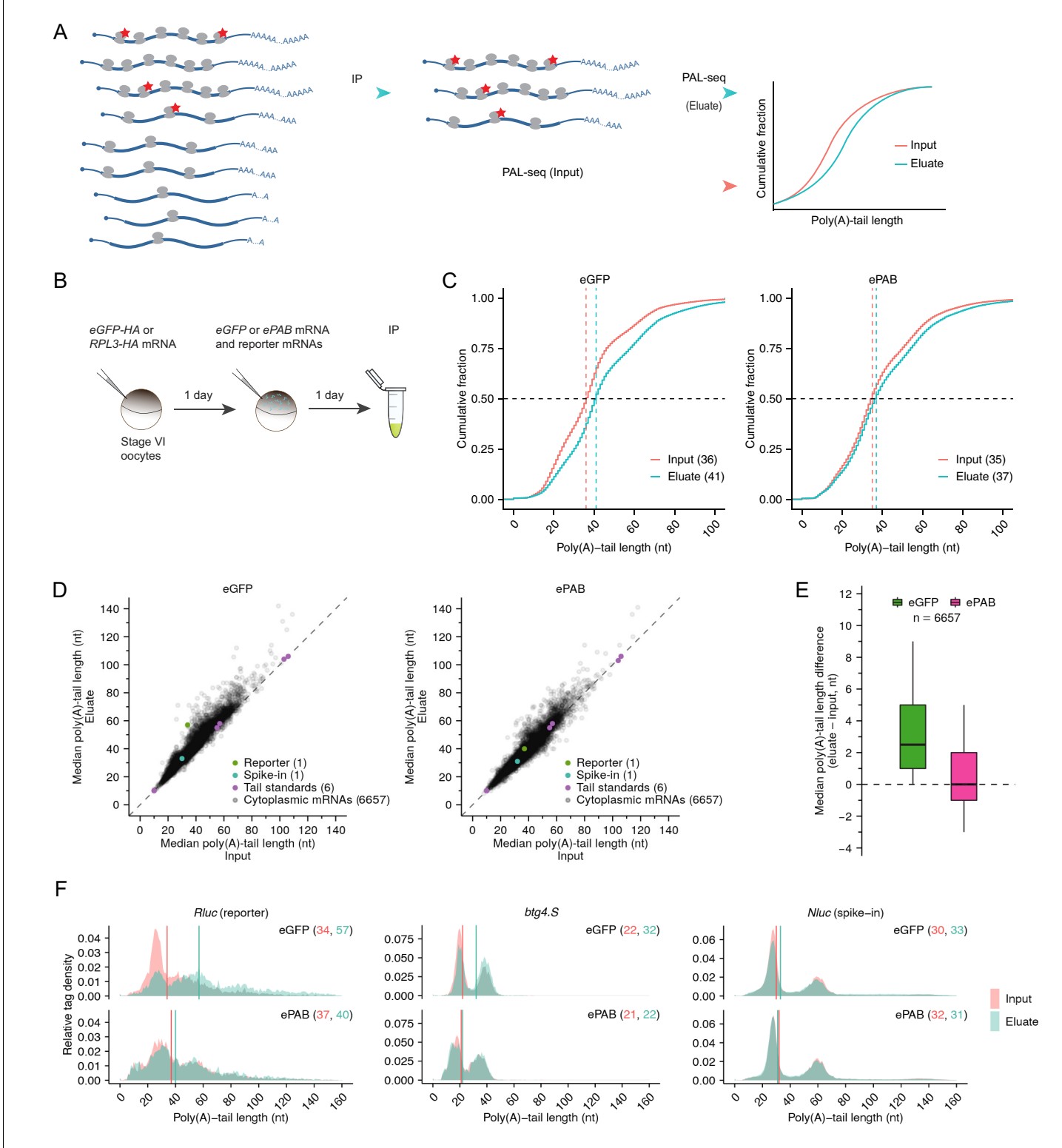

**Figure 3.** Limiting PABPC is required for intragenic coupling between poly(A)-tail length and TE. (**A**) The PAL-TRAP method for measuring intragenic effects of tail length on TE. Ribosomes are sparsely tagged (red stars) so that highly translated mRNAs are more likely to contain tagged ribosomes and thus be enriched in the immunoprecipitation (IP) eluate. Tail lengths of both input and eluate mRNAs were measured and compared for mRNAs of each gene. The depicted enrichment of long-tailed isoforms in the eluate indicates that poly(A)-tail length and TE are coupled, whereas no enrichment would indicate otherwise. (**B**) The experimental scheme of PAL-TRAP in frog oocytes. See *Figure 3—figure supplement 1B–C* for results from pulldowns using the eGFP-HA control. (**C**) Effect of overexpressing ePAB on coupling between tail length and TE, as detected after pooling PAL-TRAP

*Figure 3 continued on next page*

*Figure 3 continued*

results for mRNAs from different genes. Plotted are cumulative distributions of poly(A)-tail lengths in the PAL-TRAP input and eluate obtained after expressing either eGFP (left) or ePAB (right) in oocytes (injecting 4 fmol mRNA per oocyte). Median values are indicated (dashed lines) and listed in parentheses. (D) The effect of overexpressing ePAB on intragenic coupling between tail length and TE. Plotted for each mRNA isoform is the median poly(A)-tail length of mRNAs in the PAL-TRAP eluate compared to that in the input. Shown are results for mRNAs from oocytes either expressing eGFP or overexpressing ePAB (left and right, respectively). Each point represents an mRNA isoform with a unique 3′ end represented by ≥100 poly(A) tags in both input and eluate. Also indicated are results for (1) an *Rluc* reporter mRNA possessing a variable-length tail (reporter), which was co-injected with mRNAs expressing either eGFP or ePAB, (2) an mRNA with a variable-length tail, which was spiked into the lysate immediately before IP (spike-in), and (3) synthetic RNAs with defined tail lengths added to samples prior to library preparation (tail standards). Points for eight standards with longer tails fell outside the plot areas, as did a point representing the mRNA 3′-end from one gene (*uqcrb.S*) in the ePAB sample. (E) Summary of differences in median tail lengths observed between the eluate and the input of mRNA isoforms shown in (D). Box and whiskers indicate the 10th, 25th, 50th, 75th, and 90th percentiles. (F) Effect of overexpressing ePAB on intragenic tail-length distributions in frog oocytes. Shown are tail-length distributions of the reporter *Rluc* (left), an endogenous oocyte mRNA *btg4.S* (middle), and the spike-in mRNA *Nluc* (right) in eGFP-expressing (top) or ePAB-overexpressing (bottom) oocytes. Median tail-length values are indicated (vertical lines) and listed in parentheses. Tail-length measurements in this figure were obtained using PAL-seq v3.

The online version of this article includes the following figure supplement(s) for figure 3:

**Figure supplement 1.** Supporting data for PAL-TRAP analyses in frog oocytes.

Taken together, the PAL-TRAP results revealed intragenic coupling between poly(A)-tail length and TE for endogenous mRNAs as well as reporters in stage VI frog oocytes. Moreover, these PAL-TRAP analyses showed that as observed both in our reporter assays and in our global intergenic comparisons, substantial coupling between tail length and TE requires limiting PABPC.

## Additional conditions besides limiting PABPC are required for strong coupling

Having established the necessity of limiting PABPC for coupling, we investigated if it could also be sufficient, that is, whether limiting PABPC could confer coupling between poly(A)-tail length and TE in cells in which these features were normally uncoupled. To this end, we knocked down PABPCs in HeLa cells. Based on available mRNA-seq and mass spectrometry results (*Nagaraj et al., 2011*), PABPC1 and PABPC4 are the two major PABPC paralogs in HeLa cells, with PABPC1 four times more abundant than PABPC4 and the two together accounting for >95% of all PABPC in HeLa cells. Consistent with the idea that PABPC is not normally limiting in uncoupled systems, PABPC1 is estimated to be present in threefold excess over the poly(A) sites in HeLa cells (*Görlach et al., 1994*). To reduce PABPC to limiting levels, we used siRNAs to reduce either PABPC1 or PABPC4 alone or both PABPC1 and PABPC4 by >90% (*Figure 4—figure supplement 1A*) and examined the relationship between median tail lengths and TE. Knocking down PABPC4 alone had little impact on the coupling (*Figure 4—figure supplement 1B*), consistent with the inference that PABPC4 constitutes less than 20% of the total PABPC protein. Although correlation between median poly(A)-tail length and TE gradually increased as more PABPC was depleted, it reached an $R_s$ of only 0.18 (p<10$^{-19}$) in double-knockdown cells (*Figure 4—figure supplement 1B*), which was much weaker than that observed in frog oocytes (*Figure 2A*) and frog and fish early embryos (*Subtelny et al., 2014*). Minor $R_s$ increases were observed in other mouse and human post-embryonic cell lines in which PABPC was depleted, but strong coupling between poly(A)-tail length and TE was not established, with no $R_s$ values exceeding 0.3 (data not shown). Thus, other conditions in addition to limiting PABPC must also be met to confer strong coupling between poly(A)-length and TE.

## PABPC depletion causes premature decay of short-tailed mRNAs in mammalian cells

A striking consequence of depleting PABPC in HeLa cells was a sharp increase in median poly(A)-tail lengths, which for HeLa mRNAs increased an average of 17 and 39 nt in PABPC1- and double-knockdown cells, respectively (*Figure 4A*). Substantial changes were also observed in the distributions of global poly(A)-tail length, which showed that mRNAs with tails ranging from 10 to 50 nt were >2-fold depleted in the PABPC1-knockdown cells and that mRNAs with tails ranging from 10 to 135 nt were 2- to 20-fold depleted in the double-knockdown cells (*Figure 4B*). Similar results were obtained in NIH3T3 cells (*Figure 4—figure supplement 2A–B*). In contrast, loss of short-tailed mRNAs was

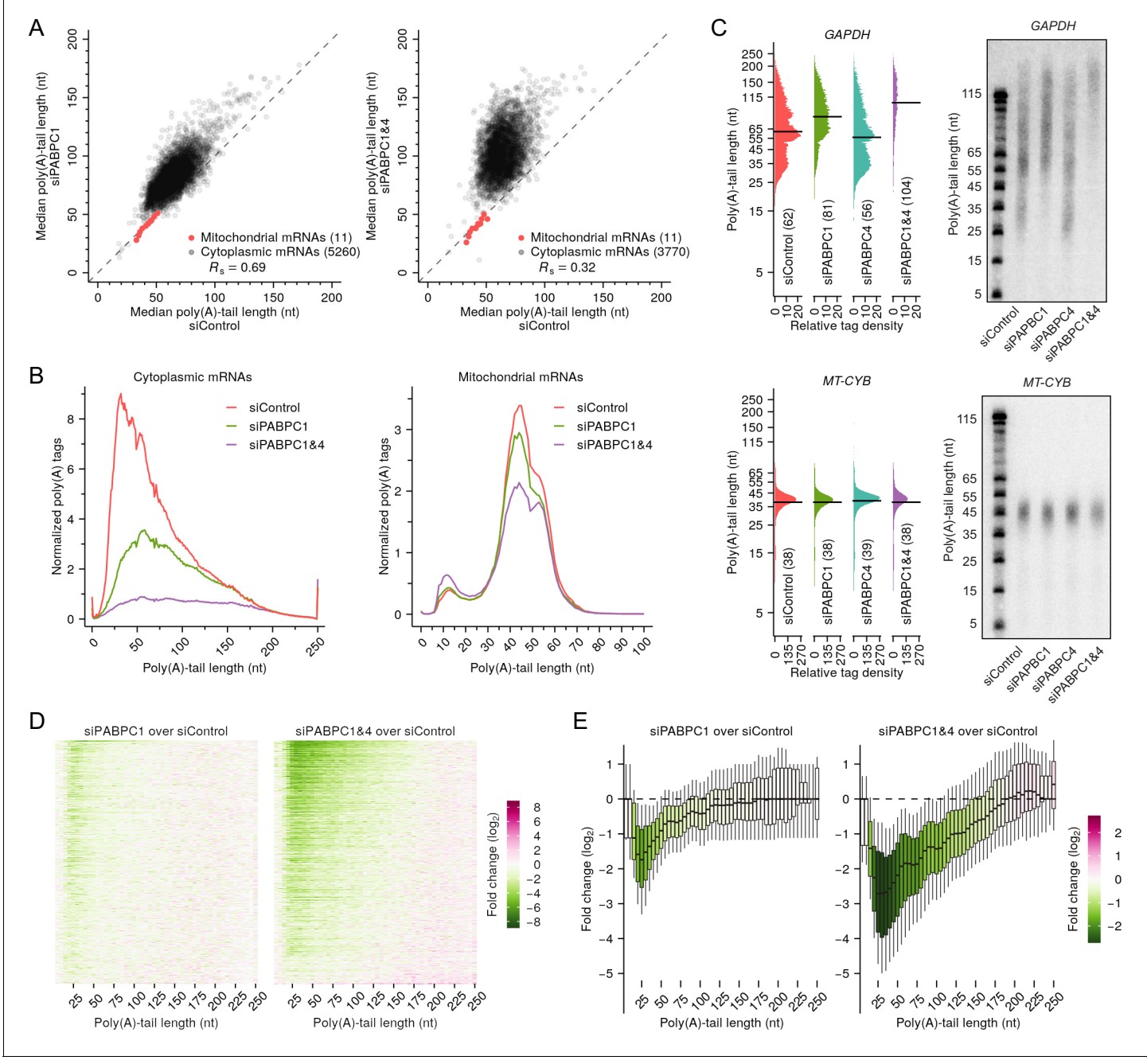

**Figure 4.** PABPC depletion causes premature decay of shorter-tail cytoplasmic mRNAs in HeLa cells. (**A**) The effect of PABPC knockdown on poly(A)-tail length. The plots compare median poly(A)-tail lengths in either PABPC1-knockdown cells (left) or PABPC1 and PABPC4 double-knockdown cells (right) to those in control cells. Results are shown for cytoplasmic mRNAs with ≥100 poly(A) tags (gray) and for mitochondrial mRNAs (red), merging data for *MT-ATP6* and *MT-ATP8* and for *MT-ND4* and *MT-ND4L*, which are bicistronic mitochondrial mRNAs. (**B**) The effect of PABPC knockdown on the abundance of mRNAs with different tail lengths. Shown are tail-length distributions of all cytoplasmic (left) and mitochondrial (right) mRNA poly(A) tags in control, PABPC1-knockdown, and double-knockdown cells. For each distribution, the abundance of tags was normalized to that of the spike-in tail-length standards. Due to depletion of tail-length calling at position 50, which was associated with a change in laser intensity at the next sequencing cycle, the values at this tail length were replaced with the average of values at tail lengths 49 and 51 nt. (**C**) The effect of PABPC knockdown on the abundance of mRNAs with different tail lengths, comparing tail-lengths measured by sequencing (left) with those observed on RNase H northern blots (right). Results are shown for a cytoplasmic mRNA *GAPDH* (top) and a mitochondrial mRNA *MT-CYB* (bottom). Relative tag density was calculated by log-transforming linear tag density using normalized poly(A) tag counts. Median tail-length values are indicated (horizontal lines) and listed in parentheses. For RNase H northern blots, a DNA oligonucleotide complimentary to the 3′-UTR was used to direct cleavage of the target mRNA by RNase H, leaving a 35-nt fragment of the 3′-UTR appended to the poly(A) tail, which was resolved on a denaturing gel and detected by a radiolabeled

*Figure 4 continued on next page*

*Figure 4 continued*

probe. Tail lengths indicated along the left side of each gel are inferred from lengths of size markers. (D) The effect of PABPC knockdown on the abundance of mRNAs with different tail lengths, extending the intragenic analysis to tail-length distributions from thousands of genes. Heat maps compare poly(A)-tag levels in PABPC1-knockdown (left) or double-knockdown (right) cells to those in control cells, after normalizing to spike-in tail-length standards, as measured using tail-length sequencing. Each row represents mRNAs from a different gene, and rows are sorted based on fold change of mRNA abundance measured using RNA-seq. Only genes with ≥100 poly(A) tags in each of two samples being compared were included in the analyses (n = 5504). Columns represent values from 5-nt tail-length bins ranging from 0 to 244 nt and a 6-nt bin ranging from 245 to 250 nt. Tile color indicates the fold change of normalized tag counts (key). (E) The effect of PABPC knockdown on the abundance of mRNAs with different tail lengths, reanalyzing data from (D) to show distributions of poly(A)-tag changes observed at different tail-lengths. Each box-whisker shows the 10th, 25th, 50th, 75th, and 90th percentile of fold changes in normalized poly(A)-tag counts observed for each tail-length bin of (D). The color of each box indicates the median value (key). Tail-length measurements in this figure were obtained using TAIL-seq.

The online version of this article includes the following source data and figure supplement(s) for figure 4:

**Figure supplement 1.** PABPC depletion is not sufficient to establish strong coupling between poly(A)-tail length and TE in HeLa cells.
**Figure supplement 2.** Support and extension of experiments showing that PABPC depletion causes premature decay of short-tailed mRNAs in HeLa cells.
**Figure supplement 3.** Measurements of mRNA half-lives in PABPC-depleted HeLa cells.
**Figure supplement 3—source data 1.** Source data for mRNA half-life values shown in *Figure 4—figure supplement 3B–C*.

not observed for mitochondria-encoded mRNAs, as expected for these mRNAs that never encounter PABPC (*Figure 4A–B*, *Figure 4—figure supplement 2A–B*), and examination of internal standards and replicates confirmed that the loss of short-tailed cytoplasmic mRNAs was not attributable to inaccurate or variable measurements (*Figure 4—figure supplement 2C–D*). Moreover, knocking down the minor isoform (PABPC4) alone did not have a similar effect (*Figure 4—figure supplement 2E*), suggesting that the tail-length changes observed for cytoplasmic mRNAs were a consequence of limiting PABPC.

We also used northern blots to examine effects of PABPC depletion on tail-length distributions of mRNAs from individual genes and found that the results corresponded well to those observed by tail-length sequencing. Both northern blots and sequencing showed strong depletion of short-tailed isoforms for a cytoplasmic mRNA (*GAPDH*) after PABPC knockdown but no substantial change in the tail-length distribution of a mitochondrial mRNA (*MT-CYB*) (*Figure 4C*). Using sequencing data to examine the intragenic tail-length distributions of cytoplasmic mRNA from each of more than five thousand other genes revealed findings resembling those observed for *GAPDH* (*Figure 4D–E*). After knocking down PABPC1, the reduction in short-tailed mRNAs was typically most severe for mRNA isoforms with tail lengths of ~25 nt, and in the double knockdown, this dip at ~25 nt became more pronounced, with reductions extending to all but the longest-tail isoforms (*Figure 4D–E*). Indeed, for more than half of the genes examined, ≥2-fold reductions extended to isoforms with tails as long as 135 nt (*Figure 4D–E*). Similar results were observed when examining tail-length distributions in the NIH3T3 dataset (*Figure 4—figure supplement 2B*), and when examining individual tail-length distributions for mRNAs of each of the top-expressed nuclear genes but not when examining those for mRNAs of mitochondrial genes (*Figure 4—figure supplement 2F–G*).

These results, which showed that mRNAs with short tails were preferentially destabilized when PABPC was depleted provided genetic loss-of-function evidence that PABPC stabilizes mRNAs in mammalian cell lines. Although genetic evidence for the role of PABPC in mRNA stability has been reported in yeast (*Caponigro and Parker, 1995*; *Coller et al., 1998*), this function had not been established in mammalian cells. Although in principle the mRNA destabilization that we observed upon PABPC-depletion might have been indirect, two lines of evidence support the conclusion that this destabilization was a direct consequence of the loss of PABPC binding to poly(A) tails. First, destabilization preferentially occurred for short-tailed mRNAs, which were expected to be the least successful at competing for binding under conditions of limiting PABPC. Second, destabilization sharply diminished at tail lengths of 10–15 nt, which corresponded to the 12 nt poly(A) sequence reported to be the minimal length that can be bound by PABPC1 with high affinity (*Kühn and Pieler, 1996*). Indeed, the modest loss observed for mRNAs with very short poly(A) tails (*Figure 4D–E*) suggested that even in control cells that had abundant PABPC, mRNAs with tails shorter than 12 nt were poorly bound by PABPCs, and thus PABPC depletion did not substantially influence their abundance.

A recent study observed similar poly(A) tail-length changes in PABPC1-depleted cells but attributed these changes to impaired deadenylation (*Yi et al., 2018*). Because PABPC can promote deadenylation in vitro (*Schäfer et al., 2019*; *Uchida et al., 2004*; *Webster et al., 2018*; *Yi et al., 2018*), it is conceivable that the loss of PABPC would slow deadenylation, thereby increasing mRNA median tail lengths, as observed in Pab1-knockout yeast (*Caponigro and Parker, 1995*). However, our analyses, which had the benefit of quantitative tail standards that enabled measurement of absolute abundance changes, revealed little added accumulation of long-tailed isoforms in PABPC-depleted cells (*Figure 4B–E*, *Figure 4—figure supplements 2B, F*), indicating that a deadenylation defect was not the major cause for the perturbed tail-length distributions. Moreover, we found that mRNA half-life values, as determined by metabolic labeling, reduced significantly when PABPC was knocked down (*Figure 4—figure supplement 3A–C*), which concurred with the conclusion that PABPC knockdown destabilized short-tailed mRNA isoforms and argued against the previous assertion that PABPC knockdown impaired deadenylation, in that impaired deadenylation would have lengthened mRNA half-lives.

Taken together, these results show that PABPC binding stabilizes mRNAs of cultured mammalian cells; if PABPC becomes limiting in these cells, the short-tailed mRNAs become destabilized, presumably because they compete less effectively for PABPC. Most importantly, the destabilization of mRNAs that competed poorly for PABPC binding helps explain why limiting PABPC was insufficient to cause strong coupling in mammalian cell lines, in that strong coupling between tail length and TE would be difficult to establish in a regulatory regime in which short-tailed mRNA molecules that lack PABPC binding are degraded rather than translated less efficiently. Thus, these results identify a second mechanistic requirement for strong coupling between tail length and TE: In addition to limiting PABPC, strong coupling requires metabolic stability of the mRNAs that compete poorly for PABPC binding.

## Terminal uridylation contributes to premature decay of short-tailed mRNAs in PABPC-depleted cells

The identification of this second requirement for strong coupling brought to the fore the question of why mRNAs that competed poorly for limiting PABPC were destabilized. To explore the possible mechanisms, we searched for perturbations that could restore stability of short-tailed mRNAs in HeLa cells undergoing PABPC knockdown, monitoring tail-length distributions of endogenous *GAPDH* using northern blots. As a positive control, expressing an siRNA-resistant PABPC1 restored stability of short-tailed species, as did frog ePAB, which further illustrated functional conservation of PABPC from different species and developmental stages (*Figure 5A*). Interestingly, a PABPC1 variant with substitutions that disrupt its interaction with eIF4G (*Chorghade et al., 2017*) also restored the stability of short-tailed species, implying that the classical closed loop is not necessary for PABPC1 to protect short-tailed mRNAs from degradation. Because PABPC has been implicated in inhibiting mRNA terminal uridylation in vitro (*Lim et al., 2014*) and in cells (*Yi et al., 2018*), and terminal uridylation has been linked to mRNA decay (*Lim et al., 2014*; *Rissland and Norbury, 2009*), we asked if terminal uridylation contributed to the loss of short-tailed mRNAs. Knocking down both TUT4 and TUT7 in PABPC-depleted cells partially restored short-tailed *GAPDH* mRNAs (*Figure 5B*). Similar results were observed in HCT116 cells, in which we tagged endogenous PABPC1 with an auxin-inducible degron (AID) and induced depletion by adding indole-3-acetic acid (IAA, a form of auxin) (*Figure 5C*; *Natsume et al., 2016*).

To examine the global rescue of short-tailed mRNAs and at the same time monitor mRNA terminal uridylation levels, we modified our tail-length sequencing protocol by including in the adaptor-ligation step a splint oligonucleotide designed to accommodate tails with a 3′ terminal U (*Eisen et al., 2020*). Knockdown of PABPC1 alone significantly increased the terminal uridylation levels across essentially all tail-length isoforms (*Figure 5D–E*), consistent with a previous report (*Yi et al., 2018*). Knockdown of PABPC4 in addition to PABPC1 further increased uridylation of mRNA isoforms with longer tail lengths (*Figure 5E*). Knockdown of TUT4 and TUT7 in PABPC-depleted cells brought terminal uridylation of all tail-length isoforms to background levels (*Figure 5D–E*) and, more importantly, preferentially rescued shorter-tail isoforms, thereby decreasing median tail lengths (*Figure 5F*). These results were consistent with those of our northern assays, and together, our results indicated that in these mammalian cells, limiting PABPC makes short- and

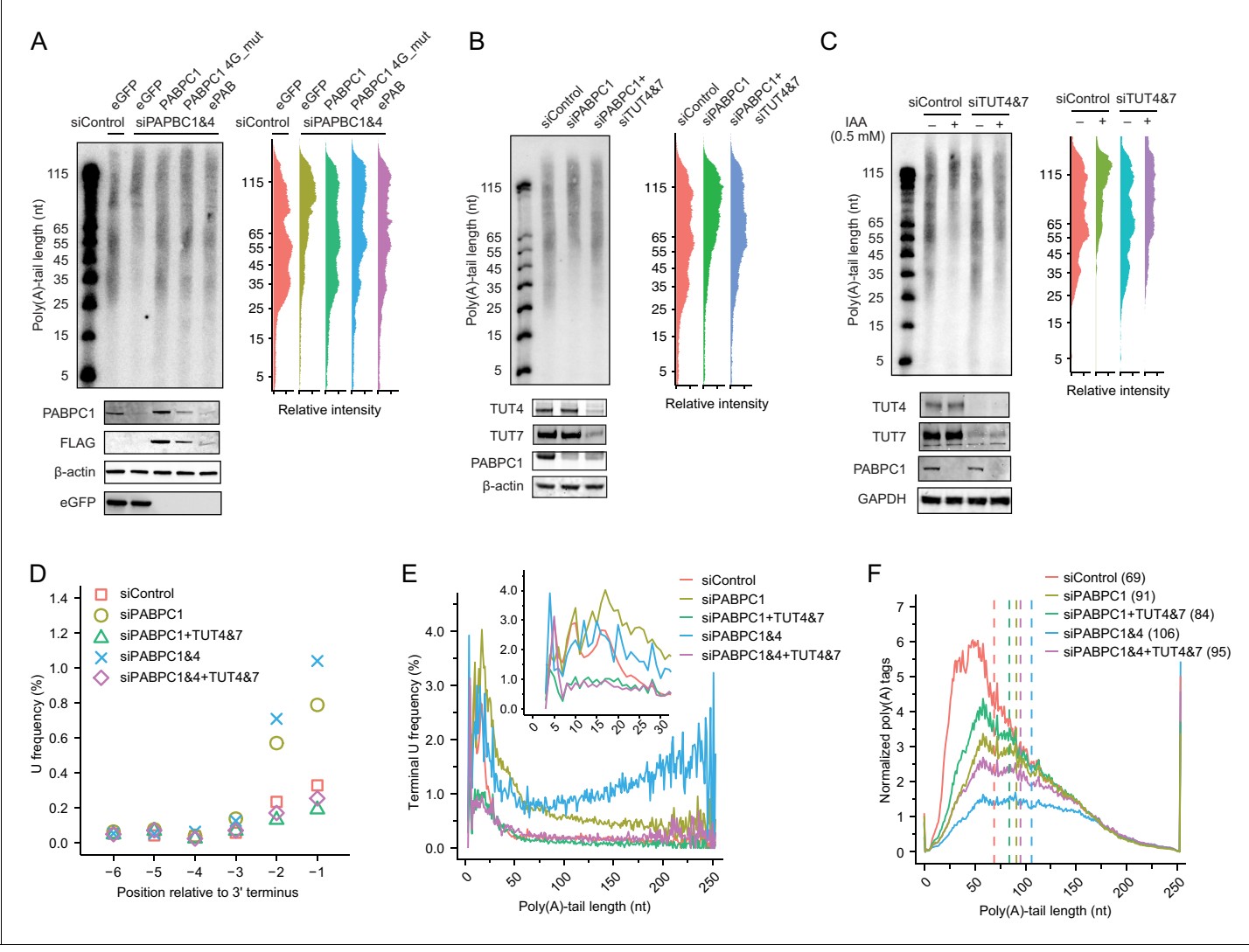

**Figure 5.** Depletion of TUT4 and TUT7 attenuates premature decay of mRNA caused by PABPC depletion. (A) Rescue of loss of shorter-tail mRNAs in PABPC-depleted HeLa cells by expressing either an siRNA-resistant human *PABPC1*, an siRNA-resistant *PABPC1* coding for a mutant that does not bind eIF4G (4G_mut), or an siRNA-resistant frog *ePAB*. At the top left is an RNase H northern blot probed for *GAPDH*, as in *Figure 4C*. At the top right are the intensity levels quantified from the blot. At the bottom is a western blot probed for the indicated proteins or the FLAG tag appended to the C terminus of each of the expressed PABPC proteins. (B) Partial rescue of loss of shorter-tail mRNAs in PABPC-depleted HeLa cells by knocking down *TUT4* and *TUT7*. At the top left is an RNase H northern blot probed for *GAPDH*, as in *Figure 4C*. At the top right are the intensity levels quantified from the blot. At the bottom is a western blot probed for the indicated proteins. (C) Partial rescue of loss of shorter-tail mRNAs in PABPC-depleted HCT116 cells by knocking down *TUT4* and *TUT7*. Endogenous PABPC1 was tagged with AID, and IAA was added for 24 hr to target the fusion protein for degradation. Otherwise, this panel is as in (B). (D) The effect of PABPC knockdown and TUT knockdown on terminal uridylation. Plotted is the fraction of uridines near the termini of cytoplasmic mRNAs in HeLa cells transfected with the indicated siRNAs, as measured using tail-length sequencing. (E) The effect of PABPC knockdown and TUT knockdown on terminal uridylation of tails with different lengths. Shown is terminal uridylation frequency of cytoplasmic mRNAs as a function of poly(A)-tail length in HeLa cells transfected with the indicated siRNAs, as measured using tail-length sequencing. The inset shows a higher-resolution view of the short-tail region. (F) Partial rescue of loss of shorter-tail cytoplasmic mRNAs in PABPC-depleted HeLa cells by knocking down *TUT4* and *TUT7*, extending the analysis to all poly(A) tags. Plotted are tail-length distributions from HeLa cells transfected with the indicated siRNAs. For each distribution, tag counts were normalized to spike-in tail-length standards. Median values are indicated (dashed lines) and listed in parentheses. Tail-length measurements were obtained using PAL-seq v3.

The online version of this article includes the following figure supplement(s) for figure 5:

**Figure supplement 1.** Minimal terminal uridylation activity in frog oocytes.

medium-tailed mRNA isoforms that poorly compete for PABPC binding more susceptible to terminal uridylation, thereby accelerating their decay.

## PABPC has little effect on TE in mammalian cell lines

Having found that destabilization of short-tailed mRNAs dampened coupling between poly(A)-tail length and TE in PABPC-depleted mammalian cells, a key question remained regarding how mRNA TE, if not influenced strongly by poly(A)-tail length, was affected in these cells. To answer this question, we conducted global profiling of HeLa cells 48 hr after siRNA transfection, a time point at which PABPC1 and PABPC4 knockdowns were substantial but secondary effects were presumably not yet too severe (*Figure 6—figure supplement 1A*). We again implemented RNA-seq and ribosome-profiling protocols that enabled absolute TE comparison by using mitochondrial mRNAs for normalization (*Rooijers et al., 2013*). Surprisingly, near-complete depletion of PABPCs had no detectable effect on global mRNA TEs (*Figure 6A–B*). Although the protein synthesis rate in double-knockdown cells was reduced to 75.7% of that observed in control cells, as measured by averaging ribosome-footprint changes observed for the 9697 analyzed genes (*Figure 6A–B*), which agreed with results of a global puromycin-based translation assay (78.7%) (*Figure 6C*), this reduced protein synthesis was fully explained by the decrease in mRNA levels, as indicated by a distribution of TE changes that centered near zero (*Figure 6A–B*). Examination of our ribosome-footprinting data revealed no upregulation of other PABPC paralogs, although the TE of *PABPC1* mRNA increased 2.5-fold (*Figure 6A*), consistent with its known autoinhibitory translational control (*Bag and Wu, 1996*). Results of polysome profiling confirmed those of our sequencing-based methods, in that the reduction in translation output, as measured by the height of polysome peaks in double-knockdown cells, was attributable to an overall decrease of mRNA levels rather than to decreased TEs that would otherwise cause a shift of mRNA distribution from heavy to light fractions (*Figure 6—figure supplement 1B*). Together, these results indicated that PABPC depletion in HeLa cells had negligible effect on mRNA TE.

The slow dynamics of siRNA-mediated knockdown, which dictated the relatively late 48 hr time point for our global measurements, might have prevented detection of any TE changes that happened earlier, before a new steady state had been reached. To examine this possibility, we monitored the dynamics of tail-length, mRNA-abundance, and translation changes soon after PABPC depletion, using the HCT116 PABPC1-AID degron cell line, in which PABPC1 was rapidly and efficiently depleted after adding IAA (85% within 30 min, >99% within 1 hr, *Figure 6D*). Because PABPC1 is the primary PABPC isoform in HCT116 cells, depletion of PABPC1 alone caused substantial destabilization of shorter-tail mRNAs 1 hr after IAA addition, and this destabilization further increased after 3 hr (*Figure 6E*). Accompanying the loss of shorter-tail mRNAs was a corresponding reduction of mRNA abundance for most genes (*Figure 6F*, *Figure 6—figure supplement 1C*). Importantly, ribosome footprints declined in lockstep with mRNA abundance, leading to median TE changes that centered near zero over the entire course of PABPC1 depletion (*Figure 6F*, *Figure 6—figure supplement 1C*). Thus, as PABPC became limiting, mRNAs that lost PABPC had no detectable reduction in TE before they were destabilized.

These data from the PABPC degron line allowed us to examine whether some coupling between tail length and TE might have occurred very soon after PABPC depletion—during the time window in which PABPC had become limiting but short-tailed mRNAs had only started to degrade. However, no coupling between tail length and TE was detected over the course of rapid PABPC depletion (*Figure 6—figure supplement 2*). Thus, in this context in which two conditions for strong coupling were satisfied (i.e. PABPC was limiting and short-tailed mRNAs were largely intact), no coupling was observed, presumably because coupling also requires a regulatory regime in which PABPC enhances translation.

These results show that in contrast to mRNAs of frog oocytes and presumably those of other coupled systems, mRNAs of HeLa and HCT116 cells do not require PABPC for efficient translation, which explains why poly(A)-tail lengths were not able to strongly influence TE after we reduced PABPC of these cells to limiting levels. Thus, these results identify a third mechanistic requirement for coupling between poly(A)-tail lengths and TE: coupling requires a regulatory regime in which PABPC affects mRNA translation.

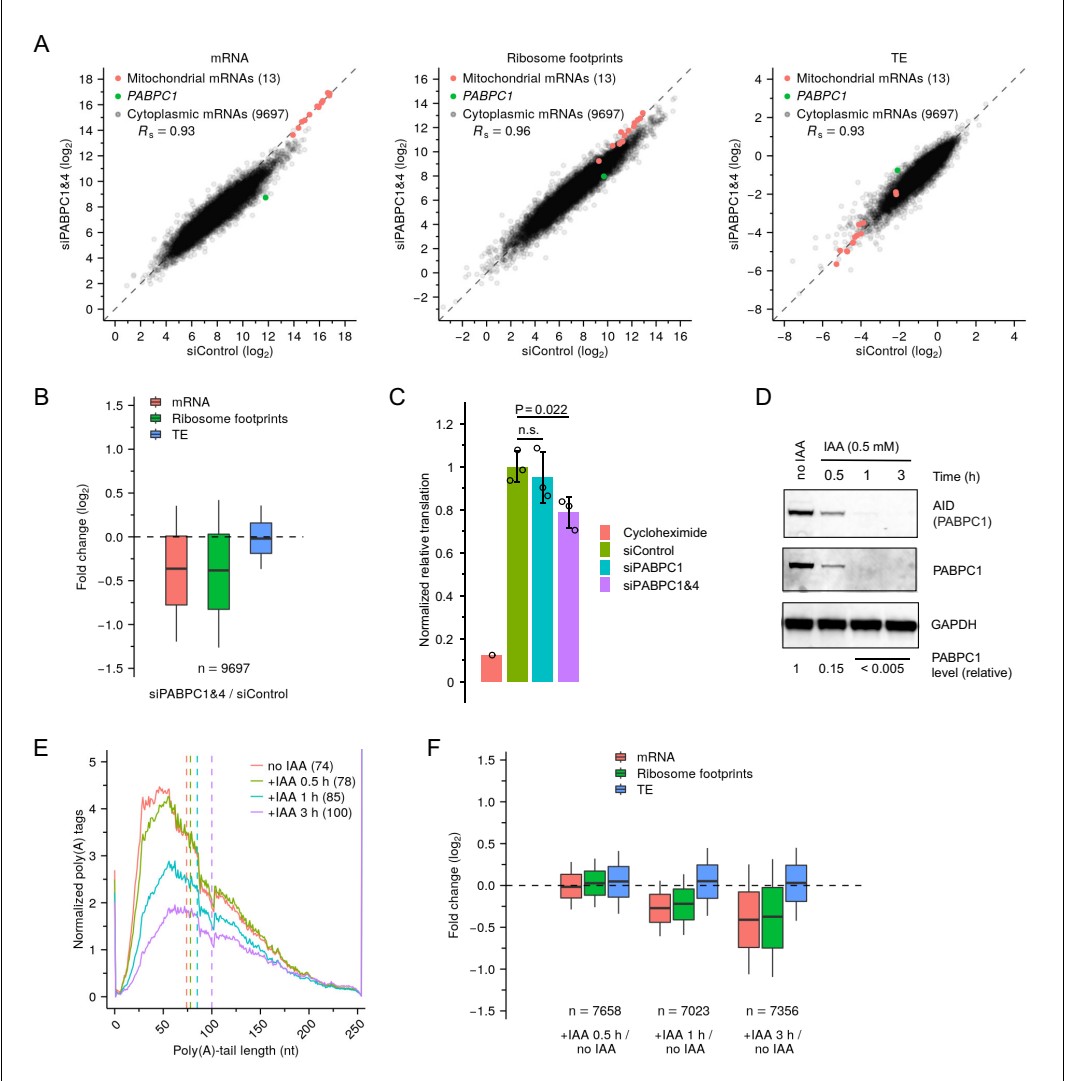

**Figure 6.** Depletion of PABPC in mammalian cell lines has minimal effect on TE. (**A**) Effect of PABPC knockdown on mRNA abundance (left), ribosome-footprint abundance (middle) and TE (right) in HeLa cells, comparing values in double-knockdown cells to those in control cells. For each gene, values for mRNA and ribosome-footprint reads per kilobase are plotted after normalizing to measurements for mitochondrial mRNAs. (**B**) Distributions of the effects of PABPC double knockdown on mRNA abundance, ribosome-footprint abundance and TE. Each box-whisker shows the 10th, 25th, 50th, 75th, and 90th percentile of the fold changes observed in (**A**). (**C**) Effect of PABPC knockdown on protein synthesis in HeLa cells. Plotted are relative levels of protein synthesis measured by pulse puromycin incorporation 48 hr after transfection with the indicated siRNAs (***Figure 6—figure supplement 1A***; error bars, standard deviation of three biological replicates; p values, one-sided *t*-tests; n.s., not significant). (**D**) Western blot showing rapid degradation of PABPC1-AID fusion protein after adding IAA to genome-engineered HCT116 cells. PABPC1 protein levels were quantified by averaging signals for AID and PABPC1, after normalizing to that for GAPDH. (**E**) Effect of PABPC1 depletion on abundance of mRNAs with different tail lengths. Shown are tail-length distributions of all poly(A) tags obtained from cytoplasmic mRNAs of HCT116 PABPC1-AID cells after treatment with IAA for the indicated time. For each distribution, the abundance of tags was normalized to that of the spike-in tail-length standards. Median values are indicated (dashed lines) and listed in parentheses. Due to depletion of 101-nt tail lengths, which was associated with a change in laser intensity at the next sequencing cycle, the values at this length were replaced with the average of values at tail lengths 100 and 102 nt. (**F**) Summary of the effects of rapid PABPC1 depletion. For each gene, at each time point after adding IAA to HCT116 PABPC1-AID cells, values for mRNA abundance, ribosome-footprint abundance and TE were each compared to the value observed in cells not treated with IAA (***Figure 6—figure supplement 1C***), and the distribution of fold changes is plotted. Each box-whisker shows the 10th, 25th, 50th, 75th, and 90th percentile. Tail-length measurements were obtained using PAL-seq v4.

The online version of this article includes the following source data and figure supplement(s) for figure 6:

**Source data 1.** Source data for values shown in ***Figure 6C*** and ***Figure 6D***.

**Figure supplement 1.** Depletion of PABPC in mammalian cell lines has minimal effect on TE.

**Figure supplement 2.** Rapid depletion of PABPC1 is not sufficient to establish detectable coupling between poly(A)-tail length and TE in HCT116 cells.

## Discussion

We find that three fundamental molecular conditions must be met for cells to use poly(A)-tail lengths to effectively regulate TE. First, PABPC must be limiting compared to the number of poly(A) sites available for binding. Under this condition, short-tailed mRNA isoforms that poorly compete for PABPC are less likely to have PABPC bound to their 3′ ends (*Figure 7*). To the extent that PABPC is not bound, these isoforms lose the translation-activating capability of PABPC observed in coupled systems. Cooperative binding of adjacent PAPBC molecules (*Lin et al., 2012*; *Schäfer et al., 2019*) would further enhance partitioning of limiting PABPC away from short-tailed mRNAs and onto longer-tailed mRNAs. When additional PABPC is introduced into a coupled system, short-tailed mRNAs benefit more from PABPC binding than long-tailed ones, which are more likely to already possess the number of PABPC molecules required for more efficient translation.

The requirement of limiting PABPC for coupling TE to poly(A)-tail length raises the question of the stoichiometry between PABPC and its sites in the poly(A) tails and whether this stoichiometry changes during the embryonic switch of gene-regulatory regimes. Our sequencing results indicated that the mRNAs of a stage VI frog oocyte have ~$2.8 \times 10^{11}$ PABPC sites (~$7 \times 10^{10}$ sites per 1 μg total RNA, ~4 μg total RNA per oocyte), which concurred with the previous estimate of $2 \sim 3 \times 10^{11}$ sites per oocyte (*Sagata et al., 1980*). The amount of PABPC (including both ePAB and PABPC1) in frog oocytes has been estimated at either $1 \times 10^{11}$ (*Cosson et al., 2002*) or $1.4 \times 10^{11}$ (*Peuchen et al., 2017*; *Smits et al., 2014*) molecules per oocyte. Our results showing that PABPC activity is limiting in frog oocytes, agreed with these estimates that imply that PABPC levels are sufficient to bind no more than half of the available PABPC sites. Poly(A)-site occupancy would be even lower if some PABPC proteins were sequestered from the mRNA pool or in an inactive form, which could be conferred by factors that either bind to or post-translationally modify PABPC to affect its ability to either bind poly(A) tails or promote translation (*Brook et al., 2012*; *Khaleghpour et al., 2001*). Until stage 15, developing frog embryos maintain the number of PABPC sites at a level resembling that of oocytes (*Sagata et al., 1980*). In contrast, the total amount of PABPC molecules increases significantly, nearly tripling by stage 12 (*Peshkin et al., 2019*)—the stage at which the coupling between tail length and TE starts to disappear (*Subtelny et al., 2014*). This increased PABPC would shift the stoichiometry toward PABPC being less limiting. Importantly, dysregulation of this tightly controlled stoichiometry not only disrupts the normal gene-regulatory regime but also can cause severe consequences during oocyte maturation and embryonic development (*Gorgoni et al., 2011*; *Wormington et al., 1996*).

In our overexpression experiment, we increased the level of PABPC1 to >6 times its endogenous level (*Figure 1—figure supplement 1E*). When considering that the ratio between endogenous

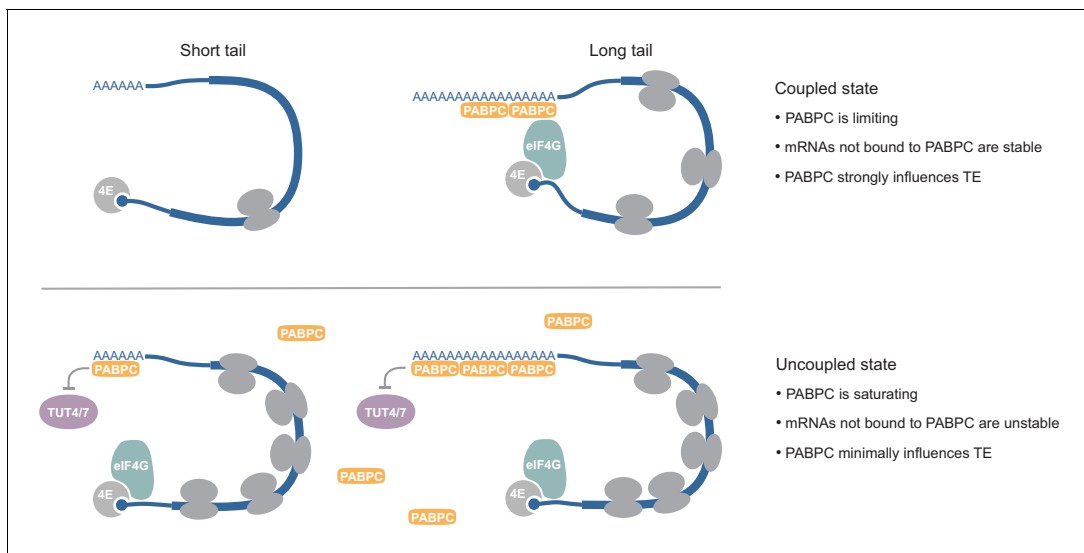

**Figure 7.** Model for coupling between poly(A)-tail length and TE, and context-dependent roles of PABPC. See text for details.

PABPC1 and ePAB is about 1:3 (*Wühr et al., 2014*), the estimated increase in overall PABPC level was >2.5-fold, which might have been sufficient to saturate the poly(A) sites. Despite this potential saturation and diminished coupling between poly(A)-tail length and TE, coupling was not completely lost (*Figures 1E*, *2A* and *3C–E*). To the extent that poly(A) sites were saturated, the residual coupling suggests that oocytes might have a mechanism for counting bound PABC molecules that enables some coupling to persist even after PABPC overexpression saturates the poly(A) sites. One way to achieve this counting would be for a critical PABPC-interacting factor to also be limiting such that long-tailed mRNAs, with their higher number of bound PABPC molecules would more effectively compete for binding to this limiting factor. A top candidate for such a factor is eIF4G. Indeed, when eIF4G was overexpressed together with PABPC1 in frog oocytes, the coupling between poly(A)-tail length and TE was further reduced (*Figure 1—figure supplement 2C*). The idea that PABPC interactions with limiting eIF4G might favor translation of long-tailed mRNAs in oocytes also helps to explain why overexpressing the PABPC1 M161A mutant was most detrimental to translation from the long-tailed reporter (*Figure 1E*).

Another way that some coupling could persist even after overexpressing saturating levels of PABPC is through sequestration of some mRNAs away from the translation machinery. One likely location for such sequestration would be germ granules, which have been implicated in regulating mRNA translation in oocytes of diverse animal species (*Voronina et al., 2011*). This mechanism might reinforce coupling between poly(A)-tail length and TE, perhaps by selectively sequestering short-tailed mRNAs from the active translation pool.

The second condition required for strong coupling between poly(A)-tail length and TE is the survival of short-tailed mRNAs under conditions in which PABPC is limiting (*Figure 7*). This condition was not met in the post-embryonic mammalian cell lines we examined. When PABPC was depleted in these uncoupled systems, many mRNA molecules, particularly short-tailed ones that presumably competed poorly for the remaining PABPC, were degraded. The preferential loss of mRNAs not bound by PABPC reduced the range of tail lengths, which correspondingly reduced the range of TEs that could potentially be imparted by coupling between tail length and TE. This reduced range also presumably reduced the ability to detect coupling, although some ability was expected to be retained, as indicated by an analysis in which data from a coupled system was sampled to match the more restricted tail-length distribution of an uncoupled system (*Subtelny et al., 2014*). More importantly, the loss of mRNAs not bound by PABPC reduced the number of PABPC-binding sites, thereby reducing the extent to which these sites were in excess over PABPC and thus reducing coupling between tail length and TE.

In oocytes and early embryos, mRNA decapping is uncoupled from deadenylation (*Gillian-Daniel et al., 1998*), which helps explain why short-tailed mRNAs survive in these systems despite our finding that they have limiting PABPC activity. We suggest two nonexclusive mechanistic explanations for this unusual decoupling of decapping from deadenylation. First, oocytes and early embryos have relatively low expression of decapping enzymes (*Ma et al., 2013*; *Peshkin et al., 2019*). Second, mRNA terminal uridylation activity is very low in frog oocytes (*Figure 5—figure supplement 1*; *Chang et al., 2018*), and it remains low throughout early embryonic development and only starts to increase dramatically after zygotic genome activation (*Chang et al., 2018*). This developmental delay of terminal uridylation might help ensure the survival of short-tailed mRNAs when PABPC is limiting and thereby enable strong coupling between poly(A)-tail length and TE in oocytes and early embryos. Later in development, short-tailed mRNAs are protected from terminal uridylation by saturating PABPC, which helps explain why deletion of TUT4 and TUT7 in mouse somatic cells has little impact on mRNA abundance (*Morgan et al., 2017*).

The third condition required for coupling between tail length and TE is that PABPC must have the ability to influence TE of bound mRNAs (*Figure 7*). In the coupled system of frog oocytes, increasing PABPC levels substantially improved TE of nearly all mRNAs (*Figure 2B–E*). In contrast, in uncoupled systems such as HeLa and HCT116 cells, severe depletion of PABPC, such that short- and medium-tailed mRNAs were markedly destabilized, had no consistent impact on mRNA TE (*Figure 6B and F*). We suspect that the differential effect of PABPC on TE observed in coupled and uncoupled systems is related to the divergent levels of basal translation initiation observed between these systems. Indeed, the overall translation measured by polysome profiles in oocytes and early embryos is much lower than that observed in either later developmental stages (*Woodland, 1974*) or post-embryonic mammalian cell lines (*Figure 3—figure supplement 1A*, *Figure 6—figure*

*supplement 1B*), which provides the opportunity for a translation-activating effect of PABPC to be more prominent in the coupled systems.

Our results showing that PABPCs, while playing a crucial role in protecting mRNA from premature decay, have minimal contribution to translation in post-embryonic mammalian cell lines might seem to contradict the well-accepted function of PABPC as a translational activator. However, many previous studies that established the role for PABPC in promoting translation were carried out in frog oocytes or early embryonic systems (*Smith et al., 2014*), where we found PABPC to globally enhance TE. Other previous studies were conducted in vitro, with mixed results: in reconstituted systems, PABPC is dispensable for translation initiation (*Mitchell et al., 2010*), and in rabbit reticulocyte lysates, PABPC has a minimal effect on translation (*Hinton et al., 2007*), whereas in some other cell extracts, PABPC activates translation (*Kahvejian et al., 2005*; *Tarun and Sachs, 1995*). Additional experiments will be required to determine whether this discrepancy between results we obtained from living cells and those obtained in some post-embryonic cell extracts are attributable to differences between cell types or to differences between cellular cytoplasm and in vitro extracts—perhaps imparted by dilution of translational components in extracts. Such studies will need to differentiate between the translation-activation and mRNA-stabilization activities of PABPC. In the meantime, it is helpful to know that PABPC stabilizes mRNAs of post-embryonic metazoan cells and that the two activities of PABPC can be context dependent, such that in frog oocytes PABPC strongly activates translation and has no effect on mRNA stability, whereas in mammalian cell lines it stabilizes mRNAs and has no detectable effect on TE.

The dual potential of PABPC in stabilizing mRNA and promoting translation bestows PABPC with distinct and context-dependent roles in regulating protein synthesis. In oocytes and early embryos, the lack of mRNA transcription and degradation leaves differential TE as the primary option for modulating protein synthesis. Our work indicates that in this context limiting PABPC proteins bind primarily to mRNAs with longer poly(A) tails and activate their translation. In contrast, in post-embryonic mammalian cell lines, where transcription, mRNA degradation, and translation are each operating at high efficiency, PABPCs protect mRNAs from pre-mature decay, enabling them to contribute the proper amount of protein during their lifetimes, but without any additional enhancement of TE. These context-dependent activities are not only crucial for understanding how coupling between tail length and TE is established in oocytes and early embryos and why it is lost later in development, they also provide mechanistic insight into the effects of the many posttranscriptional regulatory phenomena that alter poly(A)-tail lengths. For example, during miRNA-mediated repression, the Argonaute–miRNA complex binds target mRNAs and recruits factors that displace PABPC from the poly(A) tail and accelerate tail shortening (*Fabian et al., 2009*; *Giraldez et al., 2005*; *Moretti et al., 2012*; *Rissland et al., 2017*; *Wu et al., 2006*). In early zebrafish embryos, these effects are expected to disadvantage target mRNAs at competing for limited PABPC, which in this context would reduce their TE without changing their stability, thereby explaining why miRNAs primarily cause translational repression in these early embryos (*Bazzini et al., 2012*; *Subtelny et al., 2014*). However, in later embryonic development as well as in post-embryonic mammalian cells, displacement of PABPC and accelerated tail shortening would reduce PABPC binding to target mRNAs, which in this context would reduce their stability without changing their TE, thereby explaining why miRNAs primarily cause mRNA destabilization in these cells (*Bazzini et al., 2012*; *Guo et al., 2010*; *Subtelny et al., 2014*).

How PABPC promotes translation remains an enigma, although the closed-loop model has offered a sound mechanistic explanation. The interaction between eIF4G and PABPC is well characterized and provides a physical link connecting both ends of the mRNA, and some studies are able to catch a glimpse of possible circular structures of mRNAs in fixed tissues (*Christensen et al., 1987*) or in vitro (*Wells et al., 1998*). However, recent single-molecule imaging studies in mammalian cells question the widespread existence of mRNA closed-loop structures (*Adivarahan et al., 2018*; *Koch et al., 2020*). Our finding that depletion of PABPC had minimal impact on TE in mammalian cell lines supports a model in which pervasive eIF4G–PABPC-associated looping of mRNAs is generally lacking in uncoupled systems. This idea is consistent with the finding that the interaction between eIF4G and PABPC is dispensable in yeast (E.-H. *Park et al., 2011*) and HEK293 cells (*Adivarahan et al., 2018*), both of which are uncoupled systems (*Subtelny et al., 2014*). In contrast, the eIF4G–PABPC interaction is critical during frog oocyte maturation (*Wakiyama et al., 2000*), a process that relies on coupling between poly(A)-tail length and TE (*Richter and Lasko, 2011*), and

our results showed that increasing PABPC had a global effect on upregulating TE in frog oocytes. Moreover, translation of a long-tailed reporter was substantially repressed when the PABPC M161A mutant was overexpressed in frog oocytes (*Figure 1E*). Thus, the closed-loop structure, if it exists, might be more prevalent in coupled systems, such as oocytes or early embryos.

Our experiments examined only a few systems that inherently possessed either all or none of the three molecular conditions that we found to be required for strong coupling between tail length and TE, that is, limiting PABPC activity, stabilization of mRNAs lacking a bound PABPC, and PABPC-sensitive translation. Whether these conditions are broadly applicable to many other systems remains to be investigated. For example, some cells might fall in between the two extremes, processing one or two of these conditions but not all three. The results of our experiments in which we removed one of the conditions from frog oocytes or imposed one or two of the conditions in post-embryonic mammalian cells, predict that such cells that inherently fall between the two extremes have minimal if any coupling. Indeed, the concept that multiple conditions must be met before strong coupling can be established helps to explain why coupling between tail length and TE has been so infrequently detected outside the gene regulatory regime operating in oocytes and early embryos (*Subtelny et al., 2014*).

# Materials and methods

## Key resources table

| Reagent type (species) or resource | Designation | Source or reference | Identifiers | Additional information |
|---|---|---|---|---|
| Cell line (human) | HCT116 PABPC1-AID (sC152-C16) | This study | sC152-C16 | See Materials and methods for details |
| Cell line (human) | HCT116 PABPC1-AID (sC278-C2) | This study | sC278-C2 | See Materials and methods for details |
| Cell line (human) | HCT116 OsTIR | *Natsume et al., 2016* | | |
| Cell line (human) | HeLa | *Nam et al., 2014* | | |
| Cell line (human) | HeLa RPL3-3xHA (sC262-4) | This study | sC262-4 | See Materials and methods for details |
| Cell line (mouse) | NIH3T3 | *Eisen et al., 2020* | | |
| Cell line (zebrafish) | ZF4 | ATCC (CRL-2050) | | |
| Antibody | Anti-β-actin (Rabbit monoclonal) | Cell Signalling Technology (D6A8) | | 1:1000 dilution |
| Antibody | Anti-AID (Mouse monoclonal) | MBL International (M214-3) | | 1:1000 dilution |
| Antibody | Anti-ePAB (Rabbit polyclonal) | *Wilkie et al., 2005* | | 1:2000 dilution |
| Antibody | Anti-GFP (Mouse monoclonal) | Cell Signalling Technology (2955) | | 1:1000 dilution |
| Antibody | Anti-FLAG (Mouse monoclonal) | MilliporeSigma (F9291) | | 1:1000 dilution |
| Antibody | Anti-GAPDH (Mouse monoclonal) | Proteintech (60004) | | 1:1000 dilution |
| Antibody | Anti-HA (Mouse monoclonal) | Cell Signalling Technology (6E2) | | 1:1000 dilution |
| Antibody | Anti-PABPC1 (Rabbit polyclonal) | Cell Signalling Technology (4992S) | | 1:1000 dilution |
| Antibody | Anti-PABPC4 (Rabbit polyclonal) | Novus Biologicals (NB100-74594) | | 1:1000 dilution |

*Continued on next page*

*Continued*

| Reagent type (species) or resource | Designation | Source or reference | Identifiers | Additional information |
|---|---|---|---|---|
| Antibody | Anti-puromycin (Mouse monoclonal) | MilliporeSigma (MABE343) | | 1:1000 dilution |
| Antibody | Anti-RPL3 (Rabbit polyclonal) | Abcam (ab154882) | | 1:1000 dilution |
| Antibody | Anti-RPS15 (Rabbit polyclonal) | Proteintech (14957–1-AP) | | 1:1000 dilution |
| Antibody | Anti-RPS3 (Rabbit monoclonal) | Cell Signalling Technology (D50G7) | | 1:1000 dilution |
| Antibody | Anti-TUT4 (Rabbit polyclonal) | ABclonal (A5972) | | 1:1000 dilution |
| Antibody | Anti-TUT7 (Rabbit polyclonal) | Proteintech (25196–1-AP) | | 1:1000 dilution |
| Antibody | Anti-Vinculin (Mouse monoclonal) | Proteintech (66305) | | 1:1000 dilution |
| Antibody | IRDye 800CW Goat anti-Rabbit IgG (H + L) | LI-COR (926–32211) | | 1:10,000 dilution |
| Antibody | IRDye 680RD Goat anti-Mouse IgG | LI-COR (926–68070) | | 1:10,000 dilution |
| Sequence based reagent | All oligos | *Supplementary file 1* | | |
| Recombinant DNA reagent | All plasmids | *Supplementary file 2* | | |
| Other | Reference sequences | *Supplementary file 3* | | |
| Other | Masked mitochondrial pseudogenes | *Supplementary file 4* | | |
| Other | mRNA 3'-end annotations | *Supplementary file 5* | | |

## Cloning and site-directed mutagenesis

All DNA plasmids were assembled by restriction-free cloning unless explained otherwise (*Unger et al., 2010*). Site-directed mutagenesis was also carried out with this method. For plasmids used for mammalian cell transfection, human PABPC1 (from HeLa cell cDNA) and *X. laevis* ePAB (from oocyte cDNA) coding sequences were cloned into pcDNA5/FRT/TO (Thermo Fisher, V652020). For siRNA-resistant human *PABPC1*, silent mutations were introduced at D107, K108, S109, I110, D111, N112, V131, C132, D249, E250, N252, and G253. Additional substitutions were made at I110L, D111E, D117E, A121G, G139A, Y140F, T147S, and R166K to disrupt the interaction between PABPC1 and eIF4G (*Chorghade et al., 2017*). For siRNA-resistant *X. laevis* ePAB, silent mutations were introduced at C128, K129, V130, V131, T249, E250, and N252. Sequences of oligos used for mutagenesis are provided in *Supplementary file 1*. A list of plasmids used in this study is provided in *Supplementary file 2*. Plasmids and their sequence information will be available at Addgene.

## Templates for in vitro transcription

Plasmids for preparing DNA templates for in vitro transcription were assembled using the pGEM-11Zf(+) (Promega) backbone, inserting the appropriate sequence segments after the T7 promoter. HDV ribozyme sequence was obtained from the plasmid p2RZ (*Avis et al., 2012*). *X. laevis* PABPC1 (*pabpc1.S*), ePAB (*pabpc1l.L*), and RPL3 (*rpl3.L*) coding sequences were amplified from cDNA generated from *X. laevis* oocytes. *Renilla* (*Rluc*) and firefly luciferase (*Fluc*) coding sequences were obtained from pIS2 and pIS0, respectively (*Farh et al., 2005*). NanoLuc (*Nluc*) coding sequence was obtained from pNL1.1.TK (Promega). *X. laevis* β-globin 5'- and 3'-UTR sequences were obtained from pT7TS (Addgene #17091). Mouse *Malat1* 3' sequence was obtained from the Comp.25 mutant

plasmid (*Wilusz et al., 2012*). *Rluc* and *Fluc* reporters contained 5′- and 3′-UTR sequences inherited from the pGEM-11Zf(+) backbone, whereas *Nluc* reporters had the *X. laevis* β-globin 5′- and 3′-UTR sequences. Fragments containing variable poly(A) lengths were put in desired plasmids after all other DNA fragments were assembled, also using restriction-free cloning, except that C3040H competent cells (NEB) were used to amplify the assembled plasmids. Because long homopolymers tend to become shorter or get lost when plasmids are propagated in *E. coli*, individual clones were selected and checked by PCR and Sanger sequencing to confirm the desired length of each poly(A) region. These plasmid preparations were then used to generate templates for in vitro transcription without further propagation in *E. coli*.

For generating DNA templates for in vitro transcription, PCR reactions were carried out using KAPA HiFi HotStart PCR Kit, with a common 5′ primer upstream of the T7 promoter and different 3′ primers. For making RNAs ending in defined lengths of poly(A) sequence, a primer 300 ~ 600 nt downstream of the HDV cleavage site was used to facilitate separating the 5′ cleavage product from the 3′ cleavage product and the uncleaved transcript. For making RNAs not ending in defined lengths of poly(A) sequence, a primer pairing to the end of the desired 3′ UTR sequence was used. All DNA templates for in vitro transcription were purified on agarose gels using the QIAquick Gel Extraction Kit (QIAGEN). Sequences of oligos used for generating DNA templates are provided in *Supplementary file 1*.

## mRNAs for in vitro translation and oocyte injection

In vitro transcription was carried out with T7 RNA polymerase purified in house and used at 3.2 ng/µl final concentration in a standard 50 µl reaction containing 40 mM Tris pH 8.0, 21 mM $MgCl_2$, 2 mM Spermidine (Sigma), 1 mM dithiothreitol (DTT, GoldBio), 5 mM NTP (buffered ATP, UTP, CTP, GTP mix, Thermo Fisher), 0.1 units yeast inorganic pyrophosphatase (NEB), 40 units SUPERase•In (Thermo Fisher), and 1 µg DNA template. After incubation at 37°C for 2 hr, 2 units of TURBO DNase (Thermo Fisher) were added, followed by another 20 min incubation at 37°C. For constructs with the HDV ribozyme, thermal cycling was performed to enhance HDV cleavage (65°C for 90 s and 37°C for 5 min, three cycles, 50 µl of reaction per tube). Before gel loading, 2 µl 0.5M EDTA pH 8.0 and 50 µl 2x RNA Gel Loading Dye (Thermo Fisher, R0641) were added to all in vitro transcription reactions, regardless of whether the HDV cleavage step was performed or not. After incubation at 65°C for 5 min, RNAs were separated on 5% acrylamide denaturing gels (National Diagnostics, EC-829). Desired RNA bands were identified by UV-shadowing, excised, macerated and eluted in 300 mM NaCl at 23°C overnight on a rotator. The gel pieces were removed using Spin-X columns (Corning 8160), and RNAs were precipitated with isopropanol and resuspended in water for downstream reactions.

Capping of RNAs was carried out with the Vaccinia Capping System (NEB, M2080S) following the manufacturer's protocol. RNAs were then purified by phenol/chloroform extraction and ethanol precipitation. Water-resuspended RNAs were applied to Micro Bio-Spin columns (Bio-Rad, 7326250) for desalting. All RNAs were checked for integrity by visualizing on formaldehyde-agarose denaturing gels before being stored at –80°C.

## *X. laevis* oocytes for in vitro translation and injection

*X. laevis* ovaries were obtained from Nasco (LM00935). Ovaries were broken down to 2–3 cm pieces with tweezers and incubated in calcium-omitted OR-2 buffer (82.5 mM NaCl, 2.5 mM KCl, 1 mM $MgCl_2$, 1 mM $Na_2HPO_4$, 5 mM HEPES, pH 7.5) with 0.2% (w/v) collagenase A (Roche, 11088793001) on a rocker at 23°C for ~3 hr, at which point most oocytes were defolliculated. The oocytes were then washed extensively with calcium-omitted OR-2 buffer followed by complete OR-2 buffer (with 1 mM $CaCl_2$). Stage V and VI oocytes were separated from the rest with a ~ 0.8 mm diameter mesh sieve. For injection experiments, healthy stage VI oocytes were hand-picked and incubated in complete OR-2 buffer at 18°C overnight (>16 hr) for recovery before injection.

When preparing oocyte extracts, the bulk stage V and VI oocytes (500–1,000) were washed three times with ample oocyte extraction buffer (10 mM HEPES pH 7.5, 10 mM sodium acetate, 1 mM magnesium acetate and 2 mM DTT). After removing all excess buffer, oocytes were centrifuged at 20,000 g for 20 min. The middle layer containing the extract was collected with a pipette and transferred to a new tube. The collected extract was centrifuged at 20,000 g for 10 min, and the middle

layer was again collected, avoiding the top lipid layer and the bottom insoluble layer as much as possible. The extract was passed through a 0.45 µm filter, then aliquoted and stored at –80℃. The concentration of the oocyte extract was measured using the Bradford assay (Thermo Fisher, 23236) and was usually at 35–45 mg/ml.

## In vitro translation

Five hundred µl rabbit reticulocyte lysate obtained from Promega (L4151, untreated) was supplemented with 1 µl 12.5 mM Hemin (Sigma, 51280), 25 µl 1 M HEPES pH 7.5, and 2.5 µl 10 mg/ml yeast tRNA (Sigma, 10109495001). In vitro translation reactions were carried out with 5 µ l of either supplemented rabbit reticulocyte lysate or oocyte extract in total volume of 20 µl containing 10 µM amino acid mix (Promega, L4461), 0.5 units SUPERase•In, 10 mM creatine phosphate (Roche 10621714001), 200 µg/ml creatine kinase (Roche, 10127566001), 20 mM HEPES pH 7.5, 75 mM potassium acetate, 1.5 mM magnesium acetate, and 1 fmol/µl *Rluc* or *Nluc* reporter RNA with indicated poly(A) tails. When using *Rluc* reporters, 2.5 fmol/µl *Fluc* RNA with a 120 nt poly(A) region followed by a mutant mouse *Malat1* 3′ end was included in each reaction for use as a normalization control. When using *Nluc* reporters, 2.5 fmol/µl *Fluc* RNA with a histone mRNA 3′-end stem-loop was included in each reaction for use as a normalization control. When examining the effects of adding purified proteins, the reaction also included 2 µl of the indicated amount of either *X. laevis* PABPC1, eGFP, or buffer G (20 mM Tris pH 7.5, 150 mM NaCl, 5% glycerol, 5 mM DTT).

## Recombinant proteins

*X. laevis* PABPC1 (*pabpc1.S*) coding sequence was amplified from cDNA generated from *X. laevis* oocytes. eGFP was obtained from the plasmid pCS2+-eGFP (*Chen et al., 2017*). All coding sequences were cloned into pET28a (Novagen) vector by restriction-free cloning. The recombinant protein carried a hexahistidine tag at its C terminus. The plasmid was transformed into *E. coli* BL21(DE3) Star cells. After growing cells at 37℃ to an optical density ($OD_{600}$) of 0.6, expression of recombinant protein was induced with 0.5 mM isopropyl β-D-thiogalactoside (GoldBio). For purification of PABPC1, the cells continued to grow at 18℃ for 16 hr, after which they were collected by centrifugation, resuspended in 10 volumes of lysis buffer (20 mM Tris pH 7.5, 2 M NaCl, 5% (v/v) glycerol and 5 mM 2-mercaptoethanol) and lysed by sonication. Lysate was cleared by centrifugation at 25,000 g for 30 min and incubated with Ni-NTA agarose (Qiagen, 30210) at 4℃ for 1 hr (0.5 ml resin per 50 ml supernatant). The resin was washed with 20 resin volumes of buffer H (20 mM Tris pH 7.5, 2 M NaCl, 5% glycerol, 5 mM 2-mercaptoethanol and 10 mM imidazole pH 7.5) and then with 20 resin volumes of buffer L (20 mM Tris pH 7.5, 150 mM NaCl, 5% glycerol, 5 mM 2-mercaptoethanol and 20 mM imidazole pH 7.5). Proteins were eluted with buffer E (20 mM Tris pH 7.5, 150 mM NaCl, 5% glycerol, 5 mM 2-mercaptoethanol and 200 mM imidazole pH 7.5). The eluate was diluted 1:9 with buffer CL (20 mM Tris pH 7.5, 25 mM NaCl, 5% glycerol, 5 mM DTT) and loaded onto a Mono S column (GE Healthcare). Bound proteins were eluted by linear NaCl gradient with buffer CL and buffer CH (20 mM Tris pH 7.5, 500 mM NaCl, 5% glycerol, 5 mM DTT). Fractions from the desired peak were pooled, concentrated with an Amicon filter unit (Millipore, UFC805024) and applied to a Superdex 200 column (GE Healthcare) in buffer G (20 mM Tris pH 7.5, 150 mM NaCl, 5% glycerol, 5 mM DTT). Fractions from the desired peak were pooled, concentrated with an Amicon filter unit, flash-frozen in liquid nitrogen and stored at −80℃. eGFP protein was purified similarly, except all buffers had 150 mM NaCl, and the cation-exchange step was omitted.

## Oocyte injections

Healthy *X. laevis* stage VI oocytes that had recovered from defolliculation were selected for injection. When examining the effects of expressing additional PABPC, either water or the indicated amount of capped mRNA (0.25–2 pmol/µl) coding for either eGFP, frog PABPC1, frog PABPC1 (M161A), frog ePAB, or human eIF4G was injected into oocytes in a volume of 2–8 nl per oocyte (PLI-100 Plus Pico-Injector, Harvard Apparatus) at 23℃. Injected oocytes were incubated in complete OR-2 buffer at 23℃ for 20 hr, after which they were either lysed for making sequencing libraries or co-injected with an *Fluc* reporter mRNA with a histone mRNA 3′-end stem-loop (20 fmol/µl, used for normalization), and either an *Rluc* reporter mRNA with indicated poly(A)-tail length (10 fmol/µl) or an *Nluc* reporter mRNA with indicated poly(A)-tail length (10 fmol/µl) in a total volume of 2 nl per oocyte,

except in *Figure 1F*, where only 2 nl *Nluc* reporter mRNAs (10 fmol/µl) with indicated 3′ ends were injected. These oocytes were incubated in complete OR-2 buffer at 23°C for 6 hr before lysis for dual luciferase assays.

For PAL-TRAP, oocytes were injected with mRNA (1 pmol/µl) coding for either *X. laevis* RPL3-HA or eGFP-HA at a volume of 4 nl per oocyte. After incubation in complete OR-2 buffer at 23°C for 1 day, these oocytes were injected again with a mixture of mRNA coding for either eGFP or ePAB (1 pmol/µl) and a population of *Rluc* reporter mRNAs with different poly(A)-tail lengths (0, 30, 63, 98, and 120 nt, equimolar ratio; combined concentration, 166.5 fmol/µl) in a volume of 4 nl per oocyte. After incubation in complete OR-2 buffer at 23°C for another day, oocytes were collected and lysed for PAL-TRAP analysis.

## Luciferase assays

Five to 10 oocytes were lysed by vigorous shaking and pipetting in Passive Lysis Buffer (Promega, E1980), using 20 µl per oocyte. Lysates were cleared by centrifugation at 5000 g for 5 min at 4°C, after which 20 µl of each supernatant was transferred to a 96-well microplate and equilibrated to room temperature. Luciferase assays were carried out with Dual-Luciferase Reporter Assay System (Promega, E1980) in a Veritas Microplate Luminometer according to the manufacturer's protocol, regardless of whether Nluc or Rluc was used as the reporter.

## Northern blots

Total RNA (1–10 µg) was separated on an agarose-formaldehyde gel (*Mansour and Pestov, 2013*) and transferred to a nylon membrane using the Whatman Nytran SuPerCharge (SPC) TurboBlotter system (Sigma, WHA10416300). After overnight transfer, RNA was crosslinked to the membrane using a UV Stratalinker 2400 at wavelength 254 nm for a total of 1200 µJ. For probing *Rluc* and 18S RNAs, membranes were pre-incubated with ULTRAhyb Ultrasensitive Hybridization Buffer (Thermo Fisher, AM8670) at 68°C under rotation for 1 hr and then hybridized under the same conditions overnight with DNA probes, which were body-labeled with [α-$^{32}$P]dCTP (PerkinElmer) using a Random Primer DNA Labeling Kit (TaKaRa, 6045) according to the manufacturer's protocol. The templates for the labeling reactions were DNA fragments generated by PCR with gene-specific primers. After probe hybridization, membranes were washed two times (5 min each) with a low-stringency buffer (2X SSC and 0.1% SDS) at 68°C with rotation, and two times (15 min each) with high-stringency buffer (0.1X SSC and 0.1% SDS) at 68°C with rotation. For probing endogenous mRNAs, membranes were hybridized to radiolabeled gene-specific DNA oligonucleotide probes in ULTRAhyb-Oligo buffer (Thermo Fisher, AM8663) overnight at 42°C. The DNA probes were labeled with T4 PNK (NEB, M0201S) and [γ-$^{32}$P]ATP (PerkinElmer). Following hybridization, membranes were washed three times (20 min each) with a wash buffer (2X SSC and 0.5% SDS) at 42°C with rotation. The blots were then analyzed using a Typhoon FLA 7000 phosphor-imager (GE Healthcare Life Sciences). Before probing for a second mRNA on the same blot, the blots were stripped three times (20 min each) in a boiling stripping buffer (0.04% SDS) with gentle shaking and then checked for any residual radioactivity by extended phosphorimaging. Sequences of oligos used for northern blots are provided in *Supplementary file 1*.

## RNase H northern blots

Total RNA (1–10 µg) was mixed with a DNA oligo (100 pmol, *Supplementary file 1*) complimentary to a segment of the 3′-UTR in water in a volume of 15 µl. In some experiments, two DNA oligos, each complementary to a different mRNA, were added together in one reaction. After incubating the RNA–DNA mix at 85°C for 5 min, the temperature was gradually lowered to 42°C (0.1°C per s), and then 2 µl 10x Hybridase Buffer (500 mM HEPES pH 7.5, 1 M NaCl, 100 mM MgCl$_2$), 0.5 µl Hybridase (Lucigen, H39500, five units/µl) and 2.5 µl water were added to the mix. After incubation at 42°C for 20 min, nucleic acids were extracted with phenol/chloroform, precipitated with ethanol, resuspended in 1x RNA Gel Loading Dye (Thermo Fisher, R0641), and resolved on an 8% acrylamide denaturing gel (National Diagnostics, EC-829). RNA was transferred onto Hybond-NX membranes (GE Healthcare, RPN303T) using a Trans-Blot SD Semi-Dry Transfer Cell (Bio-Rad). Membranes were incubated at 60°C for 1 hr with EDC (1-ethyl-3-(3-dimethylaminopropyl)carbodiimide hydrochloride), Thermo Fisher, 22981 diluted in 0.17 M 1-methylimidazole pH 8.0 to chemically crosslink 5′

phosphates to the membrane. Blots were hybridized to radiolabeled DNA oligonucleotide probes in ULTRAhyb-Oligo buffer (Thermo Fisher, AM8663) overnight at 42°C. The DNA probes were complimentary to the 3′-UTR regions immediately adjacent to the poly(A) tails and were labeled with T4 PNK (NEB, M0201S) and [γ-$^{32}$P]ATP (PerkinElmer). Following hybridization, membranes were washed three times (20 min each) with a wash buffer (2X SSC and 0.5% SDS) at 42°C with rotation. The blots were then analyzed using a Typhoon FLA 7000 phosphor-imager. Before probing for a second RNA on the same blot, the blots were stripped as described for conventional northern blots. Sequences of oligos used for RNase H northern blots are provided in *Supplementary file 1*.

## Western blots

Samples for western blots were boiled in NuPAGE LDS Sample Buffer (Thermo Fisher, NP0007), resolved on NuPAGE 4–12% Bis-Tris protein gels (Thermo Fisher, NP0323BOX), and transferred to 0.2 μm PVDF membranes (Thermo Fisher, LC2002) with a Mini Gel Tank (Thermo Fisher, A25977), according to the manufacturer's protocol. Membranes were then blocked with 5% (w/v) non-fat dry milk in TBS buffer (20 mM Tris pH 7.5, 150 mM NaCl) for 30 min. Primary antibodies were diluted at 1:1000 in TBST buffer (20 mM Tris pH 7.5, 150 mM NaCl, 0.1% (v/v) Tween 20) and incubated with the blot at 4°C overnight. Secondary antibodies were diluted at 1:10,000 in TBST buffer and incubated with the blot at 23°C for 1 hr. Blots were analyzed using an Odyssey Clx machine (LI-COR) and the Image Studio software (LI-COR). For total protein detection, Revert 700 Total Protein Stain Kits (LI-COR, 926–11010) were used before incubation with primary antibodies, according to the manufacturer's protocol. Total protein levels were quantified with the ImageQuant TL software (GE Healthcare Life Sciences). Individual protein levels were quantified with the ImageStudio software (LI-COR, v5.2.5).

## $^{35}$S Labeling in oocytes

Groups of five oocytes were incubated with 0.5 mCi/ml EasyTag EXPRESS $^{35}$S protein labeling mix (PerkinElmer, NEG772007MC) in 100 μl complete OR-2 buffer at 23°C for 1 hr. For a control, a group of oocytes was pretreated with 100 μg/ml cycloheximide in 100 μl complete OR-2 buffer for 5 min before adding the labeling mix. After labeling, buffers were removed and oocytes were washed with 1 ml complete OR-2 buffer. After removing all residual wash buffer, 100 μl ice cold lysis buffer (20 mM HEPES pH 7.5, 100 mM KCl, 5 mM MgCl$_2$, 1% (v/v) Triton X-100, 1x Halt proteinase inhibitor cocktail (Thermo Fisher, 78429), 2 mM DTT) was added to each group, and oocytes were lysed by vigorous shaking and pipetting. Lysates were cleared by centrifuging at 5,000 g at 4°C for 5 min, and supernatants were boiled in NuPAGE LDS Sample Buffer and resolved by SDS-PAGE. Gels were dried and analyzed using the Typhoon FLA 7000 phosphor-imager and the ImageQuant TL software (GE Healthcare Life Sciences).

## PAL-TRAP

Injected oocytes (~100) were washed once with complete OR-2 buffer and three times with 1 ml ice-cold buffer RL (20 mM HEPES pH 7.5, 100 mM KCl, 5 mM MgCl$_2$, 1% (v/v) Triton X-100, 100 μg/ml cycloheximide, 20 units/ml SUPERase•In, cOmplete protease inhibitor cocktail [Sigma, 11836170001, 1 tablet per 10 ml buffer]). After removing all wash buffer, oocytes were lysed in buffer RL (10 μl/oocyte) by vigorous shaking and pipetting. Lysates were cleared by centrifugation at 5000 g at 4°C for 10 min. A portion of each supernatant (5%) was saved as the input for western analysis and RNA isolation. The remaining supernatant was incubated with anti-HA magnetic beads (Thermo Fisher, 88837) using 5 μl slurry per 100 μl supernatant. A population of *Nluc* RNAs with varied poly(A)-tail lengths (29, 63, and 139 nt, equimolar ratio) were spiked in at 5 ng per 100 μl supernatant, and the bead mixture was incubated at 4°C for 1 hr with end-to-end rotation. Beads were then immobilized using a magnetic stand, and the supernatant was removed, with a portion (5%) saved as the flow-through for western blot and RNA isolation. Beads were washed four times with 1 ml buffer RL and resuspended in 110 μl buffer RL, 10 μl of which was taken for western blot and the remainder was used for RNA isolation. RNA was isolated with 10 volumes of Tri Reagent (Thermo Fisher, AM9738) according to the manufacturer's protocol. One μg purified RNA was incubated at 37°C for 30 min with 10 units of T4 PNK (NEB, M0201S) in PNK buffer containing 20 units SUPERase•In in a 25 μl reaction to remove 2′−3′-cyclic phosphates of reporter RNAs. Two PAL-seq v3 libraries were made

from each input and eluate sample, and after monitoring reproducibility, data from each pair of replicates were merged during analyses, whereas only one library was made from the each flowthrough sample.

## Polysome gradients

Lysate preparation and centrifugation were performed the same as for ribosome profiling, except that no nuclease was added prior to centrifugation. RNA was purified using 3 volumes of Trizol LS (Thermo Fisher, 10296010) according to manufacturer's protocol and proteins were extracted by chloroform-methanol precipitation.

## Cell culture

All mammalian cells were cultured at 37°C with 5% $CO_2$. HeLa cells were obtained from the Bartel lab (*Nam et al., 2014*) and cultured in DMEM (VWR, 45000–304) with 10% FBS (TaKaRa, 631106). HeLa RPL3-3xHA (sC262-4) cells and their parental cells, a Flp-In T-Rex HeLa cell line that had an *OsTIR1* gene obtained from Andrew Holland, were cultured in DMEM with 10% FBS. HCT116 OsTIR1 cells, obtained from the Kanemaki lab (*Natsume et al., 2016*), and their derivative HCT116 PABPC1-AID (sC152-C16) cells were cultured in McCoy's 5A media (Thermo Fisher, 16600082) supplemented with 10% FBS, 2 mM L-glutamine (Thermo Fisher, 25030081). HCT116 PABPC1-AID (sC278-C2) cells were cultured as HCT116 OsTIR1 cells but supplemented with 600 µg/ml G418 (Thermo Fisher, 10131027). NIH3T3 cells were obtained from the Bartel lab (*Eisen et al., 2020*) and cultured in DMEM with 10% BCS (Sigma, 12133C). ZF4 cells were obtained from ATCC (CRL-2050) and cultured in DMEM F-12 media (ATCC, 30–2066) and 10% FBS, at 28°C with 5% $CO_2$. Mycoplasma testing was performed and no contamination was observed.

## Transfection

For siRNA transfection, cells were plated at $2 \times 10^6$ cells per 10 cm plate, cultured for 12 hr, and transfected with 15 pmol total siRNAs and 45 µl Lipofectamine RNAiMAX (Thermo Fisher, 13778500) in 1 ml Opti-MEM media (Thermo Fisher, 31985062) according to manufacturer's protocol. Cells were split 1:4 onto new plates 24 hr after transfection and then collected for analysis 48 hr (*Figure 4—figure supplement 2A-B*, *Figure 4—figure supplement 3*, *Figures 5–6*, *Figure 6—figure supplement 1*) or 72 hr (*Figure 4*, *Figure 4—figure supplement 1* and *Figure 4—figure supplement 2C–G*) after transfection. For rescue of PABPC-knockdown, cells were plated at $0.6 \times 10^6$ cells per well in 6-well plates, and transfected with 2.5 pmol total siRNAs and 7.5 µl Lipofectamine RNAiMAX in 250 µl Opti-MEM media. After 24 hr, cells were transfected with 1 µg DNA and 3 µl FuGENE HD (Promega, E2311) in 150 µl Opti-MEM media according to manufacturer's protocol, and collected for analysis 40 hr after DNA transfection. For other DNA plasmid transfections, cells were plated at $0.6 \times 10^6$ cells per well in 6-well plates, cultured for 24 hr, transfected with 1 µg DNA and 3 µl FuGENE HD (Promega, E2311) in 150 µl Opti-MEM media according to manufacturer's protocol. siRNAs were all purchased from Dharmacon, including siControl (D-001810-10-05), human siPABPC1 (L-019598-00-0005), human siPABPC4 (L-011528-01-0005), mouse siPabpc1 (L-060385-00-0005), human siTUT4 (L-021797-01-0005), and human siTUT7 (L-026009-00-0005). Plasmids and their sequence information will be available at Addgene.

## Puromycin-based translation assay

Assays were performed as described (*Schmidt et al., 2009*). Puromycin (Thermo Fisher, A1113802) was added to cell culture media at final concentration 1 µg/ml. Cells were incubated at 37°C for 10 min before harvest. For a control, cycloheximide was added to the media at final concentration 100 µg/ml 5 min before the addition of puromycin.

## IAA-induced PABPC1 degradation

AID was introduced at the C terminus of PABPC1 using Cas9-mediated genome engineering in HCT116 OsTIR cells. The 5′ and 3′ homology arms (~500 nt) flanking the stop codon of PABPC1 were amplified from genomic DNA of HCT116 OsTIR1 cells and cloned into a donor plasmid (kindly provided by Iain Cheeseman). The AID coding sequence followed by the T2A peptide and mCherry, whose sequences were obtained from pMK1221 (Addgene, #84220), was inserted immediately

before the PABPC1 stop codon. The PAM region targeted by Cas9 on the donor plasmid was mutated. The Cas9 guide RNA was cloned into pX330-BFP, as described (*McKinley and Cheeseman, 2014*). Both the donor and the Cas9 plasmids were co-transfected into HCT116 OsTIR cells at 0.5 µg each in a six-well format. After 72 hr, single cells with strong mCherry signal were sorted with flow cytometry into 96-well plates. Clones were expanded and genotyped by PCR and western blot. Only clones that were homozygous for AID integration were retained. One clonal line (sC152-C16) was used for results shown in *Figure 5C*. Six hr after siRNA transfection, 1 µg/ml doxycycline was added to induce *OsTIR1*. After another 18 hr, indole-3-acetic acid (IAA, GoldBio) was dissolved in ethanol and added to cells at a concentration of 0.5 mM. Twenty-four hr after IAA addition, cells were harvested for analysis. Sequences of oligos used for making the donor and the Cas9 plasmids are provided in *Supplementary file 1*.

IAA-induced depletion of PABPC1 in the PABPC1-AID cell line (sC152-C16) was relatively slow and generally required more than 12 hr for 90% depletion. To achieve faster depletion dynamics, more copies of *OsTIR1* were introduced. *OsTIR1* coding sequences were obtained from pBabe Puro osTIR1-9Myc (Addgene #80074) and subcloned into PiggyBac-dCas9-Tet1 (*Liu et al., 2016*) (kindly provided by Shawn Liu), replacing the dCas9 coding sequence. This plasmid was co-transfected with piggyBac Transposase expression vector (System Biosciences, PB210PA-1) into the PABPC1-AID cell line (sC152-C16). 48 hr after transfection, cells were selected in 600 µg/ml G418 for 10 d. Single clones were expanded and checked for *OsTIR1* expression. The clone with the highest *OsTIR1* expression (sC278-C2) was picked for IAA-induced PABPC1-AID degradation. To minimize background degradation, cells were incubated with 1 µg/ml doxycycline and 0.2 mM auxinole (Aobious, AOB8812), an inhibitor of OsTIR1 (*Yesbolatova et al., 2019*) for 6 hr, after which fresh media with 1 µg/ml doxycycline and 0.5 mM IAA were added. Cells were collected after 0.5, 1, and 3 hr of induction. For the non-induced condition, fresh media with 1 µg/ml doxycycline but no IAA were added, and cells were collected after 1 hr. Plasmids used for making the PABPC1-AID cell lines and their sequence information will be available at Addgene.

## Ribosome profiling and matched RNA-seq

To prepare lysate from mammalian cells, cycloheximide (CHX) was added to each plate at final concentration 100 µg/ml. Plates were immediately moved to a cold room and culture media were removed. Cells were washed twice with ice-cold PBS supplemented with 100 µg/ml CHX. After the last wash, 1 ml buffer RLL (20 mM HEPES pH 7.5, 100 mM KCl, 5 mM MgCl$_2$, 1% (v/v) Triton X-100, 100 µg/ml cycloheximide, 500 unit/ml RNasin Plus [Promega, N2615], 2 mM DTT, cOmplete protease inhibitor cocktail [Sigma, 11836170001, 1 tablet per 10 ml buffer]) was added to each 15 cm plate, and cells were scraped off and incubated on ice for 10 min. The resulting lysates were passed through a 26-gauge needle six times and cleared by centrifugation at 1300 g at 4°C for 10 min. To prepare lysate from frog oocytes, injected oocytes were washed once with complete OR-2 buffer and three times with buffer RLL. After removing all wash buffer, oocytes were lysed in buffer RLL (10 µl/oocyte) by vigorous shaking and pipetting. Lysates were cleared by centrifugation at 5000 g at 4°C for 10 min. One tenth of each cleared lysate was added to 10 volumes of Tri Reagent for total-RNA preparation, following the manufacturer's protocol. Another small portion (~10 µl) was taken for western analysis. The remainder was aliquoted, flash frozen in liquid nitrogen, and stored at –80°C.

For ribosome profiling, RNase I (Thermo Fisher, AM2294) was added to lysate, using 30 units per OD$_{260}$ unit for lysate from mammalian cells and 10 units per OD$_{260}$ unit for lysate from frog oocytes. After incubation at 23°C for 30 min, lysates were loaded onto a 10–50% sucrose gradient (20 mM HEPES pH 7.5, 100 mM KCl, 5 mM MgCl$_2$, 10–50% (w/v) sucrose, 100 µg/ml cycloheximide, 20 units/ml SUPERase•In, 2 mM DTT), centrifuged in a Beckman ultracentrifuge with SW41Ti rotor at 36,000 rpm for 2 hr, and then fractioned on a BioComp gradient fractionator. Fractions corresponding to 80S ribosomes were collected, except in experiments that generated data for *Figure 6A and B*, in which fractions corresponding to 40S, 60S, and 80S ribosomes were collected so as to avoid loss of the smaller-sized mitochondrial ribosomes (*Rooijers et al., 2013*). Collected fractions were concentrated with an Amicon filter unit (Millipore, UFC810024), and ribosome-protected RNA fragments were released in a buffer containing 20 mM HEPES pH 7.5, 100 mM KCl, 5 mM MgCl$_2$, 2 mM EDTA, 20 units/ml SUPERase•In, 2 mM DTT. Released RNA fragments were incubated with 0.2 mg/

ml Proteinase K (Thermo Fisher, AM2548) in the presence of 1% SDS at 42°C for 20 min, and then purified by phenol/chloroform extraction and isopropanol precipitation.

Sequencing libraries were prepared as described previously (*Subtelny et al., 2014*), except in experiments that generated data for *Figure 6A and B*, in which different size markers were used to isolate RNA fragments from a larger size range (20–40 nt in the first gel) so as to avoid loss of mito-chondrial ribosome-protected fragments (*Rooijers et al., 2013*).

For matched RNA-seq, rRNAs were either depleted with the RiboZero Gold Kit (Illumina, *Figure 2A*, *Figure 2—figure supplement 1A-B*, and *Figure 4—figure supplement 1B*), depleted with the NEBNext rRNA Depletion Kit (Human/Mouse/Rat, NEB, E6310L, *Figure 6* and *Figure 6—figure supplement 1–2*), or not depleted (*Figure 2B, E and F*, and *Figure 2—figure supplement 1C*). Because the RiboZero Gold Kit depletes mitochondrial mRNAs (*Figure 2—figure supplement 1C*), mitochondrial mRNAs were not used for normalization of data generated with this kit. Total RNA (with rRNA depleted or not depleted) was fragmented by incubating at 95°C for 20 min in RNA Fragmentation buffer (2 mM EDTA, 12 mM $Na_2CO_3$, 88 mM $NaHCO_3$) and then ethanol precipi-tated. RNA fragments were size-selected and sequenced in parallel with ribosome-protected frag-ments. A detailed protocol for ribosome profiling is available at http://bartellab.wi.mit.edu/protocols.html.

Sequencing was performed on an Illumina HiSeq 2500, with standard runs of either 40 or 50 cycles. Reads were trimmed at both ends to removed adapter sequences using Cutadapt (v1.18) (*Martin, 2011*) and then mapped to their respective genome reference using STAR (v2.4.2a) with the parameters '--runMode alignReads --outFilterMultimapNmax 1 --outReadsUnmapped Fastx --outFilterType BySJout --outSAMattributes All --outSAMtype BAM SortedByCoordinate'. Mapped reads were counted for each gene with htseq-count (0.11.0). For both ribosome profiling and RNA-seq, only reads that uniquely mapped to coding regions of annotated genes (excluding the first 15 codons and the last five codons) were included in downstream analyses. For TE analyses, an expression cutoff of 30 RNA-seq reads was applied for each gene, with no cutoff for ribosome-footprint reads. Normalizations of RNA-seq and ribosome-footprint reads were performed with DESeq2 (*Love et al., 2014*), considering reads for all genes passing the cutoff, except when abso-lute TE comparisons were made between two samples—in which case, RNA-seq and ribosome-foot-prints data were normalized by only considering reads from mitochondrial genes using software DESeq2 (*Love et al., 2014*). When only relative TEs were considered, TEs were manually centered at 0.

## TAIL-seq

Our implementation of TAIL-seq (*Eisen et al., 2020*) resembled that of mTAIL-seq (*Lim et al., 2016*) in that it used splint ligation to append the 3' adapter, as in PAL-seq v1 (*Subtelny et al., 2014*). Total RNA (1–30 μg) was mixed with two sets of tail-length standards (0.1 ng per 1 μg total RNA for each set) (*Subtelny et al., 2014*), and trace 5'-radiolabeled marker RNAs (*Eisen et al., 2020*), which were used to evaluate tail-length measurements and 3' ligation efficiency, respectively. These RNAs were ligated to a 3' adapter in a 64 μl reaction containing 1.5 μM 3' adapter, 1.25 μM 3' splint oligo, 0.2 mM ATP, one unit/μl RNasin Plus (Promega, N2615), 10 mM $MgCl_2$, 1x T4 RNA Ligase two reac-tion buffer (NEB, M0239S), and 0.5 units/μl T4 RNA Ligase 2 (NEB, M0239S), with the enzyme added after the other components had been mixed and incubated at 23°C for 5 min. The ligation reaction was incubated at 18°C for 18 hr. Small portions (2 μl) were removed at the start and end of the reac-tion for examining the ligation efficiency. After ligation, RNA was extracted with phenol/chloroform, precipitated with ethanol, resuspended in 10 μl water, and mixed with 90 μl 1x RNA Sequencing Buffer from the RNase T1 kit (Thermo Fisher, AM2283). After incubation at 50°C for 5 min and on ice for 5 min, 1 μl RNase T1 (one unit/μl, Thermo Fisher, AM2283) was added, and the reaction was incubated at 23°C for 15 min, followed by phenol/chloroform extraction and precipitation with the Precipitation/Inactivation Buffer in the RNase T1 kit. RNA was resuspended in 12 μl RNA Gel Load-ing Dye and RNA fragments ranging between 100 and 760 nt were purified on an 8% acrylamide denaturing gel, captured on streptavidin beads, 5' phosphorylated, and ligated to an equal mix of four phased 5' adapters as described (*Eisen et al., 2020*). cDNA was generated using SuperScript III (Thermo Fisher, 18080044) with DNA primers containing barcodes used for multiplexing, eluted from beads by base hydrolysis of RNA and resolved on 6% urea-acrylamide denaturing gels as described (*Subtelny et al., 2014*), except that cDNA fragments with sizes between 150 nt and 760

nt were selected. cDNAs were sequenced directly using a HiSeq 2500 machine in a paired-end 50-by-250 run in normal mode with a v3 kit as described (*Eisen et al., 2020*). Poly(A)-tail lengths shown in *Figure 2*, *Figure 2—figure supplement 1*, *Figure 4*, *Figure 4—figure supplement 1* and *Figure 4—figure supplement 2C–G* were measured with this method. Sequences of adapters and other oligos used for TAIL-seq are provided in *Supplementary file 1*.

## PAL-seq v3

Because TAIL-seq appears to require an 8-lane flow cell (*Eisen et al., 2020*) and thus requires many samples for efficient implementation, we stopped using this method and switched to PAL-seq. PAL-seq v3 differs from PAL-seq v2 (*Eisen et al., 2020*) in two major ways: (1) Barcodes were embedded in the 3′ adapters so that different samples could be pooled for downstream stages of library construction. (2) Sequencing was performed with cDNA rather than PCR-amplified cDNA. Each reaction was set up in a total volume of 20 µl by mixing total RNA (0.3–3 µg) with two sets of tail-length standards (0.1 ng of each set per 1 µg total RNA) (*Subtelny et al., 2014*), two splint DNA oligos (55 pmol A-splint and 2.75 pmol U-splint, which allowed polyadenylated RNAs with a terminal uridine to be more efficiently captured) and one barcoded 3′ adapter (50 pmol). Poly(A)-selected mRNA from zebrafish ZF4 cell line (0.1 ng per µg total RNA) was also added to mammalian RNA samples to enable additional assessment of tail-length measurement reproducibility. The RNA–DNA mix was incubated at 65°C for 5 min, and the temperature was gradually lowered to 16°C (0.1°C per sec) before other component were added to a volume of 30 µl containing 0.2 mM ATP, one unit/µl RNasin Plus, 10 mM MgCl$_2$, 1x T4 RNA Ligase 2 reaction buffer and 0.5 units/µl T4 RNA Ligase 2. After incubation at 16°C for 16 hr, EDTA (1 µl, 0.5 M, pH 8.0) was added to stop ligation and samples to be sequenced together (with different barcodes) were combined, phenol/chloroform extracted, and precipitated with ethanol. Ligated RNA was resuspended in 20 µl water, mixed with 180 µl RNase T1 buffer (20 mM sodium citrate pH 5.0, 1 mM EDTA, 7 M urea), heated to 50°C for 5 min, and chilled on ice for 5 min, before adding 1.6 µl RNase T1 (one unit/µl, Thermo Fisher, AM2283). After incubating the reaction at 23°C for 30 min, RNA was extracted with phenol/chloroform and precipitated using the Precipitation/Inactivation Buffer in the RNase T1 kit. Fragments between 150 and 760 nt were gel-purified, selected on beads, 5′-end phosphorylated, ligated to a 5′ adapter, and used for cDNA synthesis in the same way as in our implementation of TAIL-seq, except one of the four phased 5′ adapters was used and cDNAs with lengths between 200 and 760 nt were selected. Sequencing was performed similarly as in PAL-seq v2 (*Eisen et al., 2020*) but with some differences: (1) 1 fmol cDNA rather than PCR-amplified cDNA was used per lane. (2) A single primer was used to obtain two reads. (3) After cluster generation and sequencing-primer hybridization, and before extension of the primer through the poly(A)-tail region using the Klenow fragment, 16 dark cycles were performed in order to extend the sequencing primer past the barcode and constant regions of the 3′ adapters as well as past two nucleotides corresponding to the RNA 3′ termini. (4) 40 cycles of standard sequencing-by-synthesis were performed to yield the first sequencing read (read 1). (5) After obtaining read 1, the flow cell was stripped, the same sequencing primer was annealed, and 260 cycles of standard sequencing-by-synthesis were performed to read the barcode (six nt, which required seven cycles), a constant segment of the 3′ adapter (eight nt, which required only seven cycles because of an extra cycle in the barcode region), and the sequence of the RNA, beginning at its 3′ terminus, which revealed whether the RNA had a terminal uridine and provided information used to measure the length of the poly(A) tail. Poly(A)-tail lengths shown in *Figure 3*, *Figure 3—figure supplement 1*, *Figure 4—figure supplement 2A–B* and *Figure 5* were measured with this method. Sequences of adapters and other oligos used for PAL-Seq v3 are provided in *Supplementary file 1*.

## PAL-seq v4

We found that in PAL-seq v3, some barcoded 3′ adapters gave rise to much lower numbers of mapped reads, possibly due to low ligation efficiency. Therefore, we developed the PAL-seq v4 protocol, in which we made several changes. The 3′-ligation reaction was essentially the same, except that one 3′ adapter was used for all samples. After the ligation, each sample was processed separately rather than mixed. Ligated RNA was resuspended in 10 µl water and fragmented in a 100 µl rather than 200 µl volume, and fragments with sizes between 130 and 760 nt were then gel purified.

For reverse transcription, different primers containing the multiplexing barcode sequences as well as a region with five random nucleotides was used. After cDNA elution from the beads, half of each sample was saved at –20°C for future use. cDNA samples to be sequenced on the same lane in a flow cell were combined, ethanol-precipitated and gel-purified as in PAL-seq v3, except cDNA fragments with sizes between 190 nt and 800 nt were selected. Sequencing cluster generation was performed similarly to PAL-seq v3, using 0.3 fmol cDNA mixture for each lane. Read 1 started with 12 cycles of standard sequencing-by-synthesis that first sequenced the 5-nt random region (used to call clusters) and then sequenced the 6-nt barcode region (which required seven cycles) with a custom primer. The flow cell was stripped, a second sequencing primer annealed, and two dark cycles were performed in order to extend this primer past the two nucleotides corresponding to the RNA 3′ termini. The custom extension of the primer through the poly(A) tail region with Klenow was then performed, as in PAL-seq v2 (*Eisen et al., 2020*) and v3, followed by 40 cycles of standard sequencing-by-synthesis to complete the read 1, which was generated using two sequencing primers and had a total length of 52 nt (12 cycles before and 40 cycles after the Klenow reaction). The flow cell was then stripped, and the second sequencing primer was used for 255 cycles of standard sequencing-by-synthesis to generate read 2. Poly(A) tail lengths shown in *Figure 6E* and *Figure 6—figure supplement 2* were measured with this method. Sequences of adapters and other oligos used for PAL-Seq v4 are provided in *Supplementary file 1*.

## Genome references and gene annotations

Both human (release 25, GRCh38.p7, primary assembly) and mouse (release 10, GRCh38.p4, primary assembly) genomic sequences were downloaded from the GENCODE website. Sequences of mitochondrial pseudogenes (*Supplementary file 4*) in human and mouse genomes were masked to avoid losing mitochondrial mRNA reads due to multi-mapping. *X. laevis* genomic sequences were downloaded from the Xenbase website (v9.1 assembly, repeat masked), and scaffolds were removed so that only chromosomal sequences would be considered. The *X. laevis* mitochondrial genomic sequence was obtained from NCBI website (NC_001573.1) and appended to the *X. laevis* genome.

Both human (release 25, GRCh38.p7, main annotation) and mouse (release 10, GRCh38.p4, primary assembly) gene annotations were downloaded from the GENCODE website. *X. laevis* gene annotations (v9.1 assembly, v1.8.3.2 primary transcript) were downloaded from the Xenbase website, and only chromosomal annotations were used. *X. laevis* mitochondrial gene annotations were curated based on the information obtained from the NCBI website (NC_001573.1) and appended. For all species, annotations for protein-coding genes were extracted, and for each gene, the isoform with the longest open reading frame (ORF) was selected to represent that gene. In cases in which multiple isoforms had ORFs of the same length, the isoform with the longest transcript was selected as the reference annotation. For TAIL-seq and PAL-seq analyses, the databases were supplemented with sequences and annotations of tail-length standards, *eGFP*, *Rluc*, *Fluc*, and *Nluc*. For ribosome profiling and matched RNA-seq analyses, the databases were supplemented with coding sequences and annotations of *eGFP*, *Rluc*, *Fluc*, and *Nluc*. For RNA-seq analyses used to determine half-lives, the databases were supplemented with coding sequences and annotations of *AcGFP* (*Eisen et al., 2020*), *Rluc*, and *Fluc*. All supplemented sequences are provided in *Supplementary file 3*.

## Annotation of mRNA 3′-end isoforms

Each PAL-seq tag corresponding to an mRNA provides the site of cleavage and polyadenylation, which enabled annotation of mRNA 3′-end isoforms. Uniquely mapped tags from all PAL-seq datasets in the same cell type (either HeLa or frog oocyte) were merged. RNA-seq reads generated for comparison to ribosome profiling were similarly merged. Software HOMER (*Heinz et al., 2010*) was used to call peaks by using the merged RNA-seq data as the background and the merged PAL-seq data as the signal, with the following parameters '-style factor -o auto -strand separate -fdr 0.001 -ntagThreshold 50 -fragLength 40 -size 40 -inputFragLength 30 -center'. The peaks that intersected with annotated protein exons were retained as unique mRNA 3′ ends. The annotated 3′ ends are provided in *Supplementary file 5*.

## TAIL-seq data analysis

Phased constant sequences at the 5′ end and poly(A) sequences at the 3′ end of read one were removed with a custom script (*Eisen et al., 2020*). The trimmed sequences were mapped to reference genomes with STAR (v2.4.2a) with the parameters '–runMode alignReads `--outFilterMultimapNmax 1 --outReadsUnmapped Fastx --outFilterType BySJout --outSAMattributes All --outSAMtype BAM SortedByCoordinate`'. Mapped sequences were intersected with sequences of protein-coding genes by bedtools (v2.26.0) with the parameters 'intersect -wa -wb -bed -s', retaining only those sequences that were assigned to a single gene. Remaining clusters were then filtered, requiring at least five combined N and T bases in the first 6 nucleotides of read 2. For each library, 1% of the filtered read clusters (but no more than 50,000 and no less than 5000) were randomly picked as the training set for determining the Hidden Markov Model.

For each Illumina sequencing cluster, average intensities of each channel from position 15–50 of read one were used to normalize intensities of each channel in read 2. Then a T-signal value for each cycle of read two in each cluster was calculated by dividing normalized intensity from T channel by the sum of normalized intensities from the other three channels. If the T-signal was 0 for a specific cycle, the T-signal values from neighboring cycles (up to 10, minimum 5) were averaged to infer the value for that cycle. If a cluster had more than five cycles with a read 2 T-signal value of 0, the cluster was discarded. A five-state mixed Gaussian Hidden Markov Model (from python ghmm package) was then used to decode the sequence of states that occurred in read 2. It consisted of an initiation state (state 0), a strong poly(A)-tail state (state 1), a weak poly(A)-tail state (state 2), a weak non-poly (A)-tail state (state 3) and a strong non-poly(A)-tail state (state 4). All reads started in state 0, and all states were only allowed to go forward (from 0 to 4). The model was initialized with the following transition probability matrix (from state in row to state in column):

$$\begin{bmatrix} 0.04 & 0.93 & 0.02 & 0.01 & 0.00 \\ 0.00 & 0.94 & 0.03 & 0.02 & 0.01 \\ 0.00 & 0.00 & 0.50 & 0.40 & 0.10 \\ 0.00 & 0.00 & 0.00 & 0.60 & 0.40 \\ 0.00 & 0.00 & 0.00 & 0.00 & 1.00 \end{bmatrix}$$

The emission matrix for the mixed population one was initialized with (states in row, mean, variance and population fraction in column):

$$\begin{bmatrix} 100.00 & 1.00 & 1.00 \\ 1.50 & 1.50 & 0.95 \\ 1.50 & 1.50 & 0.75 \\ 1.50 & 1.50 & 0.50 \\ 1.50 & 1.50 & 0.25 \end{bmatrix}$$

The emission matrix for the mixed population two was initialized with (states in row, mean, variance and population fraction in column):

$$\begin{bmatrix} 0.00 & 1.00 & 1.00 \\ -1.00 & 1.50 & 0.05 \\ -1.00 & 1.50 & 0.25 \\ -1.00 & 1.50 & 0.50 \\ -1.00 & 1.50 & 0.75 \end{bmatrix}$$

After model initialization, all clusters from the training set were used to perform unsupervised training, and then the trained model was used to decode the sequence of states for all retained clusters. For each cluster, the poly(A) tail length was determined by summing the number of states in state 1 and 2. Each cluster assigned to a specific gene by read one was considered as a poly(A) tag. When evaluating tail-length distributions of mRNAs from individual genes, only results from genes with at least 100 tags were considered. Note that the HiSeq 2500 machine in high-throughput mode raises the laser intensities after 50 cycles in read 2, causing irregular T-signal at this position and lower-than-expected state transition predicted by the Hidden Markov Model. This led to a mild depletion of poly(A) tags with tail lengths called at 50 nt, but it did not affect results and conclusions made from overall tail-length distributions.

For normalization of poly(A) tags among samples, DESeq2 (*Love et al., 2014*) was used with all spike-in tail-length standards to obtain the scaling factor for each dataset. When analyzing median tail lengths, only genes with poly(A) tag counts exceeding an indicated cutoff were included in the analyses.

## PAL-seq data analysis

For PAL-seq v3, read one sequences were mapped to reference genomes with STAR (v2.4.2a) with the parameters '`--outFilterMultimapNmax 1 --outFilterType BySJout --outSAMattributes All --outSAMtype BAM SortedByCoordinate`'. Mapped sequences were intersected with annotations for protein-coding genes by bedtools (v2.26.0) with the parameters '`intersect -wa -wb -bed -S`', retaining only those tags for which the read one sequence was assigned to a single gene or to an mRNA isoform, when mRNA 3'-end annotations were used. Remaining clusters were then filtered by first removing the first 14 bases of read 2 (which corresponded to a constant region and the barcode region) and then requiring at least five combined N and T bases within the next 6 nucleotides of read 2. (This filtering step was skipped for analyses of terminal uridylation in *Figure 5* and *Figure 5—figure supplement 1*). For each library, 1% of the filtered read clusters (but no more than 50,000 and no less than 5000) were randomly picked as the training set for determining the Hidden Markov Model. The sequence of poly(A) states was determined similarly as for TAIL-seq, except a single Gaussian Hidden Markov Model was used, and the emission matrix was initialized with (states in row, mean and variance in column):

$$
\begin{bmatrix}
100.0 & 1.0 \\
2.0 & 0.5 \\
1.0 & 0.5 \\
-1.0 & 0.5 \\
-2.0 & 0.5
\end{bmatrix}
$$

For normalization of poly(A) tags among samples, DESeq2 (*Love et al., 2014*) was used with all spike-in tail-length standards to obtain the scaling factor for each dataset. When analyzing median tail lengths, only mRNA isoforms with poly(A) tag counts exceeding an indicated cutoff were included in the analyses. Data analyses for PAL-seq v4 were the same as for PAL-seq v3, except before mapping, the first 12 nt of read one were trimmed to remove the random region and the barcode region, and no nucleotides of read two were trimmed. Note that the HiSeq 2500 machine in rapid-run mode raises the laser intensities after 101 cycles in read 2 of a PAL-seq v4 run, causing irregular T-signal at this position and lower-than-expected state transition predicted by the Hidden Markov Model. This led to a mild depletion of poly(A) tags with tail lengths called at 101 nt, but it did not affect results and conclusions made from overall tail-length distributions.

## PAL-seq analysis of terminal uridylation

A variant HeLa cell line (sC262-4) was used for data shown in *Figure 5D–F* and *Figure 5—figure supplement 1*. This Flp-In T-Rex HeLa cell line had an *OsTIR1* gene knocked-in. An *RPL3-3xHA* gene was inserted using the Flp-In (Thermo Fisher) system and a single clone was picked after selection with 400 µg/ml Hygromycin B (Thermo Fisher 10687010) for 10 days. We have no reason to suspect that the unique features of this line affected any results shown. Data were processed as in PAL-seq v3, except that mapped reads were intersected with HeLa mRNA 3'-end annotations (see Annotation of mRNA 3'-end isoforms). All poly(A) tags with poly(A)-tail lengths ≥ 2 nt were used when examining the presence of U nucleotides near the ends of poly(A) tails.

## Half-life measurements

Forty-eight hr after siRNA transfection, HeLa cells were incubated with pre-warmed fresh media with 400 µM 5-ethynyl uridine (5EU, Jena Biosciences) for 1, 2, 4, and 8 hr. Cells were removed from plates by treatment with trypsin, washed once with ice cold PBS and lysed with 200 µl ice cold buffer RL. Lysates were cleared by centrifuging at 1300 g for 5 min. Supernatants were transferred to 2 ml Tri Reagent, and total RNA was prepared according to the manufacturer's protocol.

Biotinylation of 5EU-labeled RNA and purification of biotinylated RNA were performed at described (*Eisen et al., 2020*). RNA-seq libraries were prepared from purified 5EU-labeled RNA and

from total RNA with the NEXTflex Rapid Directional mRNA-seq Kit (Bioo Scientific, 5138–10). Sequencing was performed on an Illumina HiSeq 2500 with a standard 40-cycle run. Sequencing reads were mapped with STAR (v2.4.2a) with parameters '--runMode alignReads --outFilterMultimapNmax 1 --outReadsUnmapped Fastx --outFilterType BySJout --outSAMattributes All –outSAMtype BAM SortedByCoordinate'. Exon-mapped reads were counted for each gene with htseq-count (0.11.0).

Relative 5EU-labeled mRNA levels for each gene at each time point were obtained by normalizing read counts based on the counts for a 5EU-containing *GFP* RNA standard that had been spiked into each sample prior to 5EU biotinylation (*Eisen et al., 2020*). The steady-state mRNA levels of each gene were measured as the average of normalized read counts obtained from sequencing the input RNA for the 1 and 8 hr time points. A total of 987 genes with values that differed by >2-fold at these two time points were excluded from analysis. The nls package in R was used to fit the following equation for each gene:

$$y = C + log_2\left(1 - e^{-k(x-t_0)}\right),$$

where $x$ is the labelling time in hours (using 9999 h as the labeling time for the steady-state data point), $y$ is $log_2$ value of the normalized read count (level of labeled RNA), and $t_0$ is the time offset. $k$ is the decay constant, which was used to determine the half-life $t_{1/2}$ using,

$t_{1/2} = \frac{\ln(2)}{k}$, and $C$ is a coefficient determined by both $k$ and the synthesis rate $S$, such that,

$$C = log_2\left(\frac{S}{k}\right).$$

To fit $t_0$ for each condition, the value of $t_0$ was varied from 0.05 to 0.7, with an interval of 0.05, and the value that gave the smallest mean square-loss of $y$ when fitting to data from all genes was used. When fitting to results for each gene, $C$ was bound by (0, Inf), and $k$ was bound by (0, Inf). $C$ was initialized with $max(y)$, and $k$ was initialized with 0.23. Genes without a converged fit (27 of 9644, 17 of 9646, and 12 of 9728 in analysis of the siControl, siPABPC1, and siPABPC1&4 samples, respectively) were omitted from down-stream analyses. mRNA half-life values are reported in *Figure 4—figure supplement 3—source data 1*.

## Statistical analysis

Graphs were generated and statistical analyses were performed using R (*R Development Core Team, 2013*). Statistical parameters including the value of n, statistical test, and statistical significance (p value) are reported in the figures or their legends. No statistical methods were used to predetermine sample size. Statistical tests for correlations (between nonoverlapping dependent groups) were performed based on a published method (*Silver and Hittner, 2004*) using R package cocor (*Diedenhofen and Musch, 2015*). For luciferase assays of injected oocytes, each replicate refers to a group of 7–10 oocytes. For $^{35}$S labeling of injected oocytes, each replicate refers to a group of 5–8 oocytes. For the puromycin-based translation assay, each replicate refers to a separate transfection experiment.

## Data and software availability

Raw and processed data from sequencing were deposited in the GEO database (GSE166544). TAIL-seq and PAL-seq analyses were performed using a custom script written in Python 2.7 and available at https://github.com/coffeebond/PAL-seq, (copy archived at swh:1:rev: b0aa1ba99cc89ce2080069b50cd441a7718b7b03, *Xiang, 2021a*, *Xiang, 2021b*).

## Acknowledgements

We thank S Eichhorn, T Eisen, A Subtelny, S Gupta, X Wu, S McGeary, W Fang, K Lin, J Smith, J Kwasnieski and H Sive for valuable discussions; S Eichhorn, T Eisen, K Lin, and S Gupta for sharing improved methods for poly(A)-tail profiling; K McKinley, K Su, M Kanemaki, A Holland, A-B Shyu for sharing cell lines and plasmids; N Gray and M Brook for antibodies, and the Whitehead Institute Genome Technology Core for sequencing. This work is supported by NIH grant GM118135. KX is a

Cancer Research Institute Irvington Fellow supported by the Cancer Research Institute. DPB is an investigator of the Howard Hughes Medical Institute.

## Additional information

### Funding

| Funder | Grant reference number | Author |
|---|---|---|
| National Institute of General Medical Sciences | GM118135 | David P Bartel |
| Howard Hughes Medical Institute | | David P Bartel |
| Cancer Research Institute | | Kehui Xiang |

The funders had no role in study design, data collection and interpretation, or the decision to submit the work for publication.

### Author contributions
Kehui Xiang, Conceptualization, Resources, Data curation, Software, Formal analysis, Validation, Investigation, Visualization, Methodology, Writing - original draft, Project administration, Writing - review and editing; David P Bartel, Conceptualization, Resources, Supervision, Funding acquisition, Investigation, Project administration, Writing - review and editing

### Author ORCIDs
Kehui Xiang https://orcid.org/0000-0002-3770-1367
David P Bartel https://orcid.org/0000-0002-3872-2856

### Decision letter and Author response
Decision letter https://doi.org/10.7554/eLife.66493.sa1
Author response https://doi.org/10.7554/eLife.66493.sa2

## Additional files

### Supplementary files
- Supplementary file 1. Oligonucleotides used in this study.
- Supplementary file 2. Plasmids used in this study.
- Supplementary file 3. Reference sequences.
- Supplementary file 4. Masked mitochondrial pseudogenes.
- Supplementary file 5. mRNA 3′-end annotations.
- Transparent reporting form

### Data availability
Raw and processed data from sequencing were deposited in the GEO database (GSE166544). TAIL-seq and PAL-seq analyses were performed using a custom script written in Python 2.7 and available at https://github.com/coffeebond/PAL-seq (copy archived at https://archive.softwareheritage.org/swh:1:rev:b0aa1ba99cc89ce2080069b50cd441a7718b7b03).

The following dataset was generated:

| Author(s) | Year | Dataset title | Dataset URL | Database and Identifier |
|---|---|---|---|---|
| Xiang K, Bartel DP | 2021 | The molecular basis of coupling between poly(A)-tail length and translational efficiency | https://www.ncbi.nlm.nih.gov/geo/query/acc.cgi?acc=GSE166544 | NCBI Gene Expression Omnibus, GSE166544 |

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
