## [Decision Letter]

**Acceptance summary:**

This manuscript addresses a long-standing question, namely how does the poly(A) tail influence translational efficiency? It will therefore be of broad interest to readers from many areas of molecular biology including those interested in translation, mRNA stability, development and gene expression in general. The authors convincingly set out three criteria that must be met for coupling of poly(A) tail length with translation.

**Decision letter after peer review:**

Thank you for submitting your article "The molecular basis of coupling between poly(A)-tail length and translational efficiency" for consideration by *eLife*. Your article has been reviewed by 3 peer reviewers, including Timothy W Nilsen as the Reviewing Editor and Reviewer #1, and the evaluation has been overseen by James Manley as the Senior Editor.

Essential revisions:

All three reviewers were quite positive about the work. Nevertheless reviewer 3 raised a few concerns and reviewer 2 had some relatively minor suggestions. Please address these issues as thoroughly as possible. It seems like revision in the text will be sufficient.

*Reviewer #1 (Recommendations for the authors):*

This is an excellent manuscript in which Xiang and Bartel use an abundance of approaches to provide compelling evidence relevant to the coupling between poly(A)-tail length and translational efficiency. Without reiterating the results, the data are convincing and the paper is clearly written. Any concerns are too trivial to articulate. Publication of the work is recommended as is.

*Reviewer #2 (Recommendations for the authors):*

Figure 1 – there is substantial variability across different extracts. For example, the difference in translation of short vs. long tail reporter is 15.2-fold (1B), 33.8-fold (1C), and 7.4-fold (1E). What is the source of this variability? Also, some panels have error bars and others do not. Can error bars be included in all panels?

Figure 1C and 1E – How does the overexpression level of PABPC compare to endogenous levels?

Please define "poly(A) tags" the first time it is mentioned (Figure legend 2?)

Figure 2c – are the more prominent bands eGFP and PABPC1? If so, please label them.

There are several different sequencing methods used (TAIL-seq, PAL-seq v3, PAL-seq v4) – why was a given method chosen? It would be helpful to mention which method was used in the figure legends.

*Reviewer #3 (Recommendations for the authors):*

1. No error bars are presented in Figure 1A-B, why is this the case?

2. The authors conclude the loss of PABPC in cell lines results in the selective elimination of transcripts with short poly(A) tails. Can the authors exclude a role for poly(A) tail elongation in the absence of PABPC?

3. The authors conclude broad rules from a single coupled system and cell line experiments, where the depletion of PABPC has drastic consequences to the transcriptome (~70% reduction in mRNA abundance) and likely to the cell. The authors should acknowledge these limitations.

4. The authors fail to integrate previous findings in the model presented in figure 7. Morgan et al., 2017 have shown that loss of TUT4 and TUT7 in several cell systems and mouse tissues does not impact the transcriptome. The authors need to discuss these findings and integrate them into their model.

---

## [Author Response]

Reviewer #2 (Recommendations for the authors):Figure 1 – there is substantial variability across different extracts. For example, the difference in translation of short vs. long tail reporter is 15.2-fold (1B), 33.8-fold (1C), and 7.4-fold (1E). What is the source of this variability? Also, some panels have error bars and others do not. Can error bars be included in all panels?

The experiment that produced the 7.4-fold difference was performed in oocytes rather than in extracts, which might help to explain why its fold-change differed. For experiments done in extracts, one source of variability might have been the different batches of oocyte lysates. Another might have been the different buffer conditions that resulted from protein-storage buffer being added to the reactions that differed by 33.8-fold but not to those that differed by 15.2-fold.

To address the question of experimental variability and lack of error bars, these three experiments were repeated with replicates. When repeating these experiments, we used *Nluc/Rluc* reporters and an *Fluc* normalization control that had a histone mRNA 3′-end stem-loop. The new results resembled the original ones and are now shown as the (Figure 1B, C, E and Figure 1—figure supplement 1B), whereas one of the original panels has been moved to the supplement (Figure 1—figure supplement 2A).

Figure 1C and 1E – How does the overexpression level of PABPC compare to endogenous levels?

We tried to quantify the overexpression of ePAB relative its endogenous level by western, but the ePAB antibody did not seem to recognize the tagged ePAB protein we overexpressed. We observed a signal for the FLAG tag but not for overexpressed ePAB, which migrated slower on the gel. Nonetheless, we were able to detect overexpressed PABPC1 (as shown in Figure 1—figure supplement 1E) and estimated that it reached a level more than six times that of endogenous PABPC1. This information is now mentioned and discussed in our revised discussion.

Please define "poly(A) tags" the first time it is mentioned (Figure legend 2?)

We’ve added the definition in the Figure 2 legend.

Figure 2c – are the more prominent bands eGFP and PABPC1? If so, please label them.

Yes. We’ve now labeled these prominent bands.

There are several different sequencing methods used (TAIL-seq, PAL-seq v3, PAL-seq v4) – why was a given method chosen? It would be helpful to mention which method was used in the figure legends.

We’ve added reasons for choosing the sequencing methods to the methods section and state which method was used in each figure legend.

Reviewer #3 (Recommendations for the authors):1. No error bars are presented in Figure 1A-B, why is this the case?

When performing these experiments with different extracts, we observed consistent effects of adding PABPC but variable baseline ratios, presumably because of variable quality of the ovary we obtained from the vendor. To assess technical variability, we repeated the experiments in triplicate using the same batch of oocyte extract. We’ve replaced the original Figure 1A–B with the results of these new experiments, with error bars showing standard deviation for the technical replicates, and we’ve moved the original panel A to Figure 1—figure supplement 2A.

2. The authors conclude the loss of PABPC in cell lines results in the selective elimination of transcripts with short poly(A) tails. Can the authors exclude a role for poly(A) tail elongation in the absence of PABPC?

The idea that short poly(A) tails might have globally elongated with no change in mRNA stability is not supported by our findings that (1) absolute levels of long-tailed mRNAs (determined using normalization to spike-in standards) remained constant and did not increase, (2) mRNA half-lives did decrease, and (3) TUT4/7, which have known roles in mRNA degradation but not tail lengthening, were required for the loss of short tails.

3. The authors conclude broad rules from a single coupled system and cell line experiments, where the depletion of PABPC has drastic consequences to the transcriptome (~70% reduction in mRNA abundance) and likely to the cell. The authors should acknowledge these limitations.

The reason we created the PABPC-AID fusion and examined the effects of PAPBC depletion before large changes to the transcriptome could occur was to avoid the secondary effects of concern to the referee. Nonetheless, we now acknowledge the limitation of examining only a few systems in the last paragraph of our discussion.

4. The authors fail to integrate previous findings in the model presented in figure 7. Morgan et al., 2017 have shown that loss of TUT4 and TUT7 in several cell systems and mouse tissues does not impact the transcriptome. The authors need to discuss these findings and integrate them into their model.

We mention the results of Morgan et al. and integrate them into our model in our revised discussion. The finding that loss of TUT4 and TUT7 in somatic mouse cells does not influence mRNA abundance is consistent with our hypothesis that PABPC is not limiting in these cells. According to our model, in these cells in which PABPC is normally not limiting, PABPC coats the mRNA poly(A) tails thereby protecting them from TUT4 and TUT7.